# Connecting Jensen–Shannon and Kullback-Leibler Divergences: A New Bound for Representation Learning

**Reuben Dorent**
Inria
reuben.dorent@inria.fr

**Polina Golland**
MIT
polina@csail.mit.edu

**William Wells III**
Harvard, MIT
sw@bwh.harvard.edu

## Abstract

Mutual Information (MI) is a fundamental measure of statistical dependence widely used in representation learning. While direct optimization of MI via its definition as a Kullback-Leibler divergence (KLD) is often intractable, many recent methods have instead maximized alternative dependence measures, most notably, the Jensen-Shannon divergence (JSD) between joint and product of marginal distributions via discriminative losses. However, the connection between these surrogate objectives and MI remains poorly understood. In this work, we bridge this gap by deriving a new, tight, and tractable lower bound on KLD as a function of JSD in the general case. By specializing this bound to joint and marginal distributions, we demonstrate that maximizing the JSD-based information increases a guaranteed lower bound on mutual information. Furthermore, we revisit the practical implementation of JSD-based objectives and observe that minimizing the cross-entropy loss of a binary classifier trained to distinguish joint from marginal pairs recovers a known variational lower bound on the JSD. Extensive experiments demonstrate that our lower bound is tight when applied to MI estimation. We compared our lower bound to state-of-the-art neural estimators of variational lower bound across a range of established reference scenarios. Our lower bound estimator consistently provides a stable, low-variance estimate of a tight lower bound on MI. We also demonstrate its practical usefulness in the context of the Information Bottleneck framework. Taken together, our results provide new theoretical justifications and strong empirical evidence for using discriminative learning in MI-based representation learning.

## 1 Introduction

Estimating and optimizing Mutual Information (MI) is central to many representation learning frameworks [4, 18, 8]. The MI between two random variables $U$ and $V$ is defined as the Kullback-Leibler Divergence (KLD) between their joint distribution and the product of their marginals,

$$\mathrm{I}\left[U; V\right] \doteq \mathrm{D_{KL}}[p_{UV} \,||\, p_U \otimes p_V] \,. \tag{1}$$

Mutual information serves as a natural measure of dependence between $U$ and $V$. In representation learning, a common goal is to learn compact yet informative representations by maximizing the MI between encoded variables.

A popular class of methods for estimating MI involves learning a discriminator that distinguishes between samples from the joint distribution and those from the product of marginals. However, because the MI objective depends on a discriminator trained through a separate optimization problem, maximizing MI leads to a bilevel optimization problem, which is intractable in practice. To bypass this challenge, tractable Variational Lower Bounds (VLBs) [3, 16, 20, 24] could be used, such as the

39th Conference on Neural Information Processing Systems (NeurIPS 2025).

Donsker–Varadhan bound [3], to optimize a lower bound of the MI using data-estimated expectations. Indeed, KLD belongs to the family of $f$-divergences, which have standard VLBs. However, VLBs are often unstable and underperforming in the context of representation learning [8, 18, 13].

In practice, many successful methods optimize alternative $f$-divergences, such as the Jensen-Shannon divergence (JSD) [8] between the joint $p_{UV}$ and the product of marginals $p_U \otimes p_V$, without knowing about the relationship between KLD and JSD. We refer to this measure as ($I_{JS}$),

$$I_{JS}[U; V] \doteq D_{JS}[p_{UV} \,||\, p_U \otimes p_V] \,. \tag{2}$$

While optimizing JSD has been shown to reach competitive results, the reason why optimizing a JSD-based dependence measure would increase standard MI, defined via KLD, remains unclear.

In this work, we bridge this gap by showing that maximizing JSD is equivalent to maximizing a lower bound on MI. Our main contribution is a new and optimal lower bound on KLD as a function of JSD,

$$\Xi\left(D_{JS}[p \,||\, q]\right) \leq D_{KL}[p \,||\, q] \tag{3}$$

where $\Xi(\cdot)$ is a strictly increasing function, analytically described below, that resembles a Logit function. In particular, this result shows that maximizing $I_{JS}$ also increases a guaranteed lower bound on mutual information.

Furthermore, we revisit the practical optimization of $I_{JS}$. We show that the $f$-divergence VLB for JSD is equivalent to optimizing the cross-entropy (CE) loss of a specific discriminator. This discriminator is trained to distinguish between true pairs of data and false pairs generated by scrambling the true pairs, a connection originally noted in the GAN paper [6]:

$$\log 2 - \mathcal{L}_{CE} \leq D_{JS}[p \,||\, q] \,. \tag{4}$$

While this bound is equivalent to the standard $f$-divergence VLB for JSD, we provide a new information-theoretic derivation that highlights the discriminator perspective. Beyond offering a more intuitive interpretation, this formulation also makes explicit the gap due to suboptimal discriminators.

Together, these two results imply that minimizing the CE loss of a discriminator trained to separate joint from marginal samples leads to an increase in a lower bound on the mutual information:

$$\Xi\left(\log 2 - \mathcal{L}_{CE}\right) \;\leq\; \Xi\left(I_{JS}[U; V]\right) \;\leq\; I[U; V] \,. \tag{5}$$

This work therefore provides a principled theoretical justification for existing representation learning approaches that maximize discriminative JSD-based losses to increase mutual information, and it introduces a theoretically grounded alternative to commonly used variational lower bounds.

The rest of the paper is organized as follows. Section 2 reviews related work. Section 3 introduces the joint range of $f$-divergences. Section 4 presents our new lower bound relating JSD and KLD, along with a bound connecting cross-entropy loss and JSD. Section 5 provides empirical evidence of the bound's tightness and its utility for representation learning tasks. Section 6 concludes this work.

## 2   Related work

**Estimating Mutual Information:** MI is a core concept in information theory, but its estimation remains notoriously difficult, particularly in high-dimensional or continuous settings with limited data. Traditional estimators can be broadly categorized as parametric or non-parametric. Parametric methods rely on strong distributional assumptions (e.g., Gaussianity), which often do not hold in practice. Non-parametric approaches, such as histogram binning, $k$-nearest neighbors [10], and kernel density estimation [15, 11], are assumption-free but suffer from poor scalability with respect to data dimensionality and sample size, limiting their applicability in modern machine learning contexts.

To overcome these limitations, recent work has introduced variational lower bounds (VLBs) on MI that leverage the expressiveness of deep neural networks. A key example is MINE [3], which leverages the Donsker-Varadhan (DV) representation of the KLD to train a neural network to maximize a lower bound on MI. A related estimator, NWJ [16], arises from the dual representation of $f$-divergences and shares similar objectives. However, theoretical work by McAllester & Stratos [14] shows that such estimators are fundamentally upper-bounded by $\log(n)$, where $n$ is the number of data samples. Moreover, both MINE and NWJ exhibit high variance, in part due to unstable estimation of the

partition function. To tackle this, Poole *et al.* [20] introduce hybrid VLB methods that generalize existing methods to trade off bias and variance. SMILE [24] further improves upon MINE by clipping the density ratio, which helps reducing the estimator variance. It also introduces a consistency check that exposes weaknesses in existing VLB estimators, especially in high MI regimes, and highlights that optimizing VLBs does not necessarily imply an accurate estimation of a lower bound of MI.

Another lower bound approach is InfoNCE, introduced via contrastive predictive coding (CPC) [18]. InfoNCE considers MI estimation as a multi-way classification problem, producing stable and low-variance estimates. However, this estimator is biased and upper-bounded by $\log(b)$, where $b$ is the batch size, which limits its ability to capture high MI values.

Beyond VLB methods, another class of estimators decouples the estimation and optimization stages. NJEE [22] extends theoretical work by McAllester & Stratos [14] and estimates MI through entropy subtraction using neural density models. In contrast, DEMI [13] and PCM [26] reformulate MI estimation as a binary classification task: a model is trained to distinguish paired from unpaired samples, and discriminator outputs are then mapped to MI estimates. Interestingly, SMILE [24] can also be interpreted within this discriminator-based framework. As pointed out in [12], its implementation[1] relies on maximizing an estimator of the JSD [17], with MI subsequently estimated from the outputs of a clipped discriminator. Overall, by disentangling optimization and estimation using a two-step procedure, these methods alleviate some instability issues of VLBs. More recently, Letizia *et al.* [12] extended this setup by incorporating general $f$-divergences and empirically observed that the JSD (referred as GAN-DIME) provides the best tradeoff between bias and variance, outperforming alternatives based on KLD or Hellinger distance with a relatively low-variance MI estimation. While these two-step estimators are currently among the most accurate MI estimates in practice, they require retraining whenever the underlying joint distribution changes. This limitation is particularly restrictive in representation learning, where the latent variables evolve during optimization. As a result, these methods cannot be reused as plug-in MI estimators during training. In contrast, our bound can be directly optimized via JSD, using a single discriminator, enabling stable, end-to-end training.

**Maximizing Mutual Information in representation learning:** Several recent methods use MI maximization as a principle for representation learning via Information Bottleneck [25]. VLB-based methods such as MINE and NWJ have also been applied to representation learning, although their practical limitations, such as high variance, make them less attractive compared to contrastive approaches [8, 18]. Alternatively, contrastive predictive coding (CPC) [18] and its InfoNCE objective have shown strong empirical results across domains such as audio, vision, and NLP. However, since the estimation is upper-bounded by $\log(b)$, where $b$ is the batch size, it requires a large batch size, which may be impractical computationally.

Another popular approach is Deep InfoMax from Hjelm *et al.* [8], which shifts the focus from strictly maximizing MI to maximizing other measures of statistical dependence, particularly the JSD-based objective $I_{JS}$. Unlike KLD-based bounds used in MINE or NWJ, the JSD is symmetric and bounded, often resulting in more stable optimization, and, unlike CPC and MINE, does not require a large batch size. Deep InfoMax leverages a JSD-based MI estimator introduced by Nowozin *et al.* [17]. To support the use of JSD-based optimization objective, Hjelm *et al.* [8] empirically show that for randomly sampled discrete sparse distributions, $I_{JS}$ correlates well with true MI, suggesting it can act as a meaningful proxy. In this work, we go further and provide a theoretical justification: we show that maximizing $I_{JS}$ provably increases a lower bound on the true mutual information.

**Jensen-Shannon divergence in GANs:** The link between JSD and discriminator training has its roots in Generative Adversarial Networks (GANs), where Goodfellow *et al.* [6] showed that, under an optimal discriminator, the GAN objective minimizes an affine transformation of the JSD between the real and generated data distributions. This was later generalized in the $f$-GAN framework [17], where GAN training corresponds to a variational lower bound on a general $f$-divergence, estimated via a learned discriminator. Chen *et al.* [4] proposed InfoGAN, which incorporates MI maximization within the GAN framework using a variational approximation originally introduced by Barber & Agakov [2]. Rather than estimating the MI between joint and marginal distributions directly, InfoGAN targets the MI between a subset of latent variables and the generator's output, approximated using an auxiliary network and a variational lower bound from Barber & Agakov [2]. While InfoGAN brings information-theoretic principles into adversarial training, it does so in a generative setting with a specific structural assumption on the latent space.

---

[1] https://github.com/ermongroup/smile-mi-estimator

## 3 Joint range of $f$-divergences

To establish a novel lower bound on mutual information, we build on the concept of the *joint range* of $f$-divergences, which we summarize here. While this concept has been covered in classical information theory [27, 7], to our knowledge, it has not been utilized in the context of representation learning or mutual information estimation.

Let $p$ and $q$ be two probability distributions defined on the measurable space $(\mathcal{X}, \mathcal{F})$. For any convex function $f : (0, \infty) \to \mathbb{R}$ such that $f(1) = 0$, the $f$-divergence between $p$ and $q$ is defined as:

$$D_f(p\|q) \doteq \mathbb{E}_q \left[ f \left( \frac{dp}{dq} \right) \right]. \tag{6}$$

Common examples include the Kullback-Leibler divergence, total variation distance, and Jensen-Shannon divergence, each corresponding to a different choice of $f$.

Given two such convex functions $f$ and $g$, the joint range of the pair $(D_f, D_g)$ is the set of all achievable values:

$$\mathcal{R}_{f,g} = \{(D_f(p\|q), D_g(p\|q)) : p, q \in \mathcal{P}\}, \tag{7}$$

where $\mathcal{P}$ denotes the set of all probability distributions on some measurable space.

In addition, the joint range over all $k$-dimensional categorical distributions $[k]$ is defined as:

$$\mathcal{R}_{k;f,g} = \{(D_f(p\|q), D_g(p\|q)) : p, q \text{ are probability measures over } [k]\}. \tag{8}$$

The joint range $\mathcal{R}_{f,g}$ has several useful properties; it is convex, and the lower envelope of the joint range is the optimal lower bound relating the two divergences.

Remarkably, to characterize the entire joint range $\mathcal{R}_{f,g}$, it is sufficient to restrict to binary distributions, i.e. Bernoulli distributions [7, 27].

**Theorem 3.1** (Harremoës-Vajda, 2011 [7])**.**

$$\mathcal{R}_{f,g} = co\left(\mathcal{R}_{2;f,g}\right) \tag{9}$$

*where co denotes the convex hull.*

## 4 Maximizing discriminator cross-entropy increases a lower bound on Mutual Information

In this section, we use $f$-divergence properties, including the previously introduced joint range, to demonstrate that training a discriminator to differentiate samples from the joint and product of the marginal distributions leads to maximizing a lower bound of the Mutual Information.

### 4.1 General lower bound on Kullback–Leibler divergence by Jensen–Shannon divergence

We introduce here a new optimal lower bound on the KLD in terms of the JSD, obtained by analyzing the joint range of the two divergences.

The KLD ($D_{KL}$) and JSD ($D_{JS}$) between two probability distributions $p$ and $q$ are defined as

$$D_{KL}[p \,\|\, q] \doteq \mathbb{E}_p \left[ \log \frac{p}{q} \right], \quad D_{JS}[p \,\|\, q] \doteq \frac{1}{2} D_{KL}[p \,\|\, m] + \frac{1}{2} D_{KL}[q \,\|\, m], \tag{10}$$

where $m \doteq \frac{1}{2}(p + q)$ is a mixture distribution of $p$ and $q$.

To derive the tightest lower bound of KLD in terms of JSD, we analyze the joint range of the two divergences, i.e., the set of all possible pairs $(x, y) \in [0, \log 2) \times \mathbb{R}^+$ such that:

$$x = D_{JS}[p \,\|\, q] \quad \text{and} \quad y = D_{KL}[p \,\|\, q] \tag{11}$$

for *all* pairs of distributions $p, q$ that are compatible with the divergences.

Following Theorem 3.1 by Harremoës & Vajda [7], this joint range can be fully characterized by restricting $p, q$ to be pairs of Bernoulli distributions compatible with JSD and KLD,

$$x = D_{JS}[B(\mu) \,\|\, B(\nu)] \quad \text{and} \quad y = D_{KL}[B(\mu) \,\|\, B(\nu)], \tag{12}$$

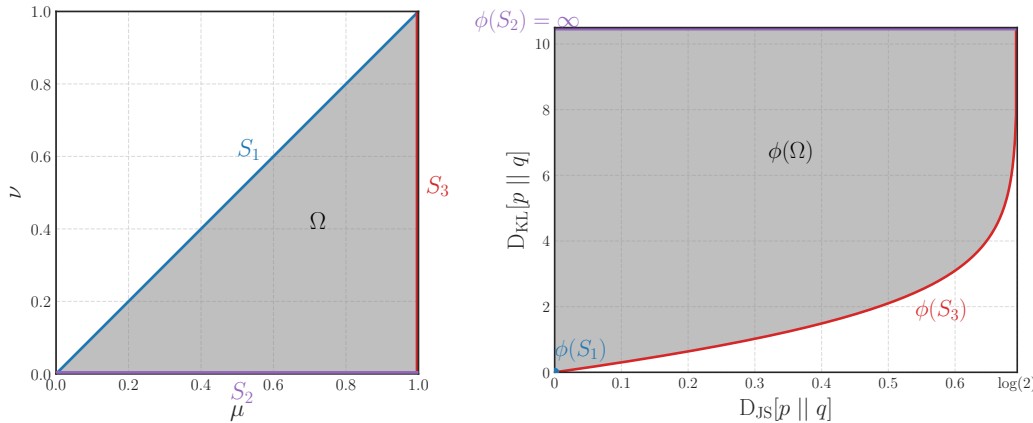

Figure 1: Left: Parameter space of two Bernoulli distributions. Right: Image (in gray) of the mapping $\phi$, showing the joint range of the Jensen–Shannon and the Kullback–Leibler divergences. The red curve is our new lower bound.

for $(\mu, \nu) \in [0, 1] \times (0, 1) \cup \{(0, 0), (1, 1)\}$, where $B(\theta)$ is a Bernoulli distribution with parameter $\theta$.

We are therefore interested in the image of the mapping $\phi$ defined as:

$$\phi : (\mu, \nu) \mapsto (\mathrm{D_{JS}}[B(\mu) \mid\mid B(\nu)], \mathrm{D_{KL}}[B(\mu) \mid\mid B(\nu)]) \tag{13}$$

from the unit square $[0, 1] \times (0, 1) \cup \{(0, 0), (1, 1)\}$ into $\mathbb{R}^2$.

Exploiting the symmetry of the divergences:

$$\mathrm{D_{KL}}[B(\mu) \mid\mid B(\nu)] = \mathrm{D_{KL}}[B(1 - \mu) \mid\mid B(1 - \nu)], \tag{14}$$

$$\mathrm{D_{JS}}[B(\mu) \mid\mid B(\nu)] = \mathrm{D_{JS}}[B(1 - \mu) \mid\mid B(1 - \nu)], \tag{15}$$

we restrict our analysis to the image $\phi(\Omega)$ of the lower triangle $\Omega = \{(\mu, \nu) \in [0, 1] \times (0, 1] \mid \nu \leq \mu\}$.

Let us analyze the behavior of $\phi$ on the three edges of the triangle as shown in Fig. 1.

1. **Diagonal** ($S_1 : \mu = \nu$):
$$\mathrm{D_{KL}}[B(\mu) \mid\mid B(\mu)] = 0, \quad \mathrm{D_{JS}}[B(\mu) \mid\mid B(\mu)] = 0.$$
Thus, all points on the diagonal map to the origin: $\phi(\mu, \mu) = (0, 0)$.

2. **Lower Edge** ($S_2 : \nu \to 0$):
$$\mathrm{D_{KL}}[B(\mu) \mid\mid B(0)] \to +\infty \text{ for any } \mu > 0.$$
The JSD remains finite, so points on this edge map to $(0, \log 2) \times \{\infty\}$.

3. **Right edge** ($S_3 : \mu = 1$): For this segment, we can obtain:
$$\mathrm{D_{KL}}[B(1) \mid\mid B(\nu)] = -\log \nu, \tag{16}$$

$$\mathrm{D_{JS}}[B(1) \mid\mid B(\nu)] = \log 2 - \frac{1}{2}\left(\log(1 + \nu) - \nu \log\left(\frac{\nu}{1 + \nu}\right)\right). \tag{17}$$

As $\nu \to 0$, $\mathrm{D_{KL}}[B(1) \mid\mid B(\nu)] \to +\infty$, while $\mathrm{D_{JS}}[B(1) \mid\mid B(\nu)]$ increases to its maximum, i.e. $\log 2$.

We now present the main result, which shows that maximizing the JSD increases a lower bound on the KLD and that this lower bound corresponds to the image of $S_3$.

**Theorem 4.1.** *The optimal lower bound (i.e., equality is achieved for a family of distributions) on Kullback-Leibler divergence established by the Jensen-Shannon divergence is:*

$$\Xi\left(\mathrm{D_{JS}}[p \mid\mid q]\right) \leq \mathrm{D_{KL}}[p \mid\mid q] \tag{18}$$

*where $\Xi : [0, \log 2) \mapsto \mathbb{R}^+$ is a strictly increasing function defined implicitly by its inverse,*

$$\Xi^{-1} : y \mapsto \mathrm{D_{JS}}[B(1) \mid\mid B(\exp(-y))], \tag{19}$$

*where $B(\exp(-y))$ is a Bernoulli distribution with parameter $\exp(-y) \in [0, 1/2)$. $\Xi^{-1}$ has a closed-form expression related to Eq. 17 and defined in Appendix (41).*

In particular, this result holds when $p$ and $q$ are respectively the joint and the product of the marginals, leading to a novel lower bound in information theory between $\mathrm{I_{JS}}$ and MI. Theorem 4.1's proof relies on a conjectured functional property, supported by extensive numerical evidence (see Appendix B.1).

**Approximation:** Although $\Xi$ does not have a closed-form expression, it can be approximated pointwise using numerical solvers (example in C.2). Alternatively, we introduce a smooth differentiable approximation using the Logit function, $\mathbb{L}(x) \doteq \log \frac{x}{1-x}$ (details in Appendix C.3):

$$\Xi(x) \approx 1.15 * \mathbb{L}\left(.5\left(\frac{x}{\log 2} + 1.0\right)\right) . \tag{20}$$

### 4.2 $f$-divergence lower bound on JSD from discriminator cross-entropy

Having established that the JS-based information ($\mathrm{I_{JS}}$) provides a lower bound on the MI, we now show that, in turn, the $\mathrm{I_{JS}}$ can be lower bounded using the expected cross-entropy (CE) loss of a discriminator trained to distinguish between dependent and independent pairs. Therefore, optimizing a binary cross-entropy classifier effectively increases a lower bound on MI via the inequality chain.

We adopt a standard binary classification setup to estimate the JSD between the joint distribution $p_{UV}$ and the product of its marginals $p_U \otimes p_V$. Consider the mixture model:

$$(U, V) \mid Z = 1 \sim p_{UV}, \quad (U, V) \mid Z = 0 \sim p_U \otimes p_V, \quad \text{with } Z \sim B(1/2). \tag{21}$$

This defines a joint distribution over $(u, v, z)$, denoted $\tilde{p}(u, v, z)$, with marginal over $(u, v)$:

$$\tilde{p}(u, v) = m(u, v) \doteq \tfrac{1}{2} p_{UV}(u, v) + \tfrac{1}{2} p_U(u) p_V(v). \tag{22}$$

Since the JSD is a $f$-divergence, it admits a variational representation [17, 27], expressible as:

$$\mathrm{I_{JS}}[U; V] = \mathrm{D_{JS}}[p_{UV} \mid\mid p_U \otimes p_V] = \tfrac{1}{2} \max_t \left[ \underset{p_{UV}}{\mathbb{E}}[t] - \underset{p_U \otimes p_V}{\mathbb{E}}[-\log(2 - \exp(t))] \right] \tag{23}$$

where $t : \mathcal{U} \times \mathcal{V} \to (-\infty, \log 2)$ is a variational function.

Now, define the following reparameterization,

$$t(u, v) \doteq \log(2 q_\theta(z = 1 \mid u, v)), \tag{24}$$

where $q_\theta(z = 1 \mid u, v)$ is meant to approximate $\tilde{p}(z = 1 \mid u, v)$, the optimal classifier.

Substituting Eq. (24) into Eq. (23) (details in B.2), we obtain the following lower bound:

$$\mathrm{I_{JS}}[U; V] = \frac{1}{2} \max_\theta \left[ \log 4 + \underset{p_{UV}}{\mathbb{E}}[\log q_\theta(z = 1|u, v)] + \underset{p_U \otimes p_V}{\mathbb{E}}[\log(1 - q_\theta(z = 1|u, v))] \right] \tag{25}$$

Thus, we obtain

$$\mathrm{I_{JS}}[U; V] \geq \log 2 - \min_\theta \mathcal{L}_{\mathrm{CE}}(\theta) , \tag{26}$$

where $\mathcal{L}_{\mathrm{CE}}(\theta)$ is the expected binary cross-entropy loss between $q_\theta$ and the true posterior.

This bound is tight in the non-parametric limit (infinite data and model capacity). An equivalent result was shown in the context of GAN discriminators [6] and can also be derived from [23] via the reparameterization $d(u, v) \doteq \mathbb{L}(q_\theta(z=1 \mid u, v))$. In Appendix B.3, we present an alternative proof that quantifies the gap between the JSD lower bound and the one obtained from cross-entropy.

Combining (26) and (18), we obtain our new VLB $I_{CE}(\theta)$ on MI as a function of a discriminator CE:

$$I_{CE}(\theta) \doteq \Xi\left(\log 2 - \mathcal{L}_{CE}(\theta)\right) \leq \Xi(\mathrm{I_{JS}}[U; V]) \leq \mathrm{I}[U; V] . \tag{27}$$

### 4.3 Estimating Mutual Information via discriminators

While we introduced a new VLB on MI (Eq. (27)) that can be estimated from data, we now show that it also enables the construction of an estimator of the true MI, as detailed below. Beginning with the joint model of Eq. (21), we derive $\tilde{p}(z|u, v)$ and then $\frac{\tilde{p}(z=1|u,v)}{\tilde{p}(z=0|u,v)}$. From this, it is easy to show that

$$\mathrm{I}[U; V] = \underset{p_{UV}}{\mathbb{E}}[\mathbb{L}(\tilde{p}(z = 1 \mid u, v))] , \tag{28}$$

where $\mathbb{L}(\cdot)$ is the Logit transform. Details can be found in Appendix B.4.

This result suggests a two-step estimation procedure: 1) train a discriminator $q_\theta$ to distinguish between joint and marginal samples; 2) estimate the MI by plugging the approximate $q_\theta$ into Eq. (28), and approximate expectations by data averages. This two-step strategy is equivalent to GAN-DIME [12].

# 5 Experiments

In this section, we evaluate the tightness of our proposed MI lower bound, derived from the JSD, and compare it to widely used variational MI estimators. Our evaluation is three-fold. First, we provide an empirical analysis of the lower bound in the discrete setting, where both MI and JSD can be computed exactly. This allows us to assess the tightness of the bound independently of neural approximation errors. Second, we assess the quality of our bound as a variational objective function in neural estimation tasks, using synthetic benchmarks introduced in prior work [20, 24, 12]. These include Gaussian and non-Gaussian settings designed to challenge both the bias and variance of MI estimators across regimes of increasing complexity and mutual dependence. Finally, we demonstrate the practical usefulness of optimizing JSD as a proxy of MI in the context of the Information Bottleneck (IB) framework.

## 5.1 Tightness of the JSD lower bound in the discrete setting

We begin by analyzing the proposed MI lower bound in a controlled discrete setup where both the true MI and the JSD between the joint distribution and the product of marginals are tractable. This allows us to assess whether our general optimal lower bound, defined for arbitrary distributions, remains tight when specialized to the MI setting, i.e. restricted to joint and their product of marginals.

We construct a parameterized family of joint distributions $P_{UV}^{(\alpha)} \in \mathbb{R}^{k \times k}$ that interpolate between the independent and perfectly dependent cases. The marginal distributions are uniform categorical $P_U = P_V = \frac{1}{k}\mathbf{1}_k$, where $k$ is the number of categories. The joint $P_{UV}^{(\alpha)}$ distribution is defined as:

$$P_{UV}^{(\alpha)} = (1 - \alpha)P_U \otimes P_V + \alpha \operatorname{diag}(P_U), \tag{29}$$

where $\alpha \in [0, 1]$ controls the strength of dependence. For each value of $\alpha$, we compute the ground-truth MI $D_{\mathrm{KL}}[P_{UV}^{(\alpha)} \parallel P_U \otimes P_V]$ and the JSD $D_{\mathrm{JS}}[P_{UV}^{(\alpha)} \parallel P_U \otimes P_V]$. We then compare $\mathrm{I}[U; V]$ with our bound $\Xi(\mathrm{I}_{\mathrm{JS}}[\mathrm{U}; \mathrm{V}])$, where $\Xi$ is the transformation defined in (19).

Figure 2 shows the resulting trajectories as $\alpha$ varies from 0 (independent variables) to 1 (maximally dependent variables) across set sizes $k \in \{2, 3, \ldots, 500\}$. For any given JSD value $x$, we observe that there exists a discrete distribution with JS-based information $\mathrm{I}_{\mathrm{JS}} \approx x$ whose true mutual information nearly coincides with the lower bound $\Xi(x)$. This empirically demonstrates the tightness of our proposed bound. This result is particularly remarkable as it suggests that the optimal lower bound between JSD and KLD introduced in Theorem 4.1, originally derived for arbitrary pairs of distributions, remains tight when restricted to mutual information estimation, i.e., when comparing joint distributions to the product of their marginals. In Appendix D, we further show that there exists an *infinite family* of discrete joint–marginal distribution pairs that lie exactly on our bound, thereby demonstrating that the proposed lower bound remains tight even under this restriction. This implies that our JSD-based lower bound offers a tight proxy when maximizing mutual information.

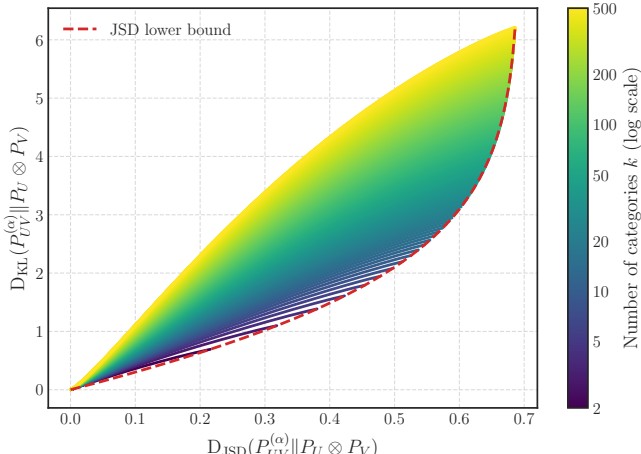

Figure 2: Mutual information and its JSD-based lower bound $\Xi(\mathrm{I}_{\mathrm{JS}})$ for a parameterized family of discrete joint distributions with known MI and JSD, varying in dependence strength ($\alpha$) and number of categories ($k$). The presence of MIs near the lower bound across settings empirically demonstrates the tightness of our JSD-based estimate.

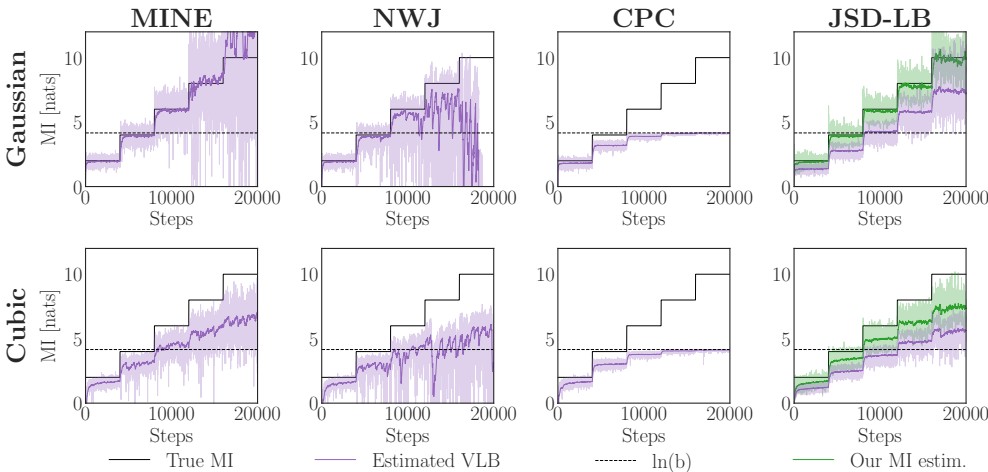

Figure 3: Staircase MI estimation comparison for $d = 5$ and batch size $b = 64$. The Gaussian case is reported in the top row, while the cubic case is shown in the bottom row.

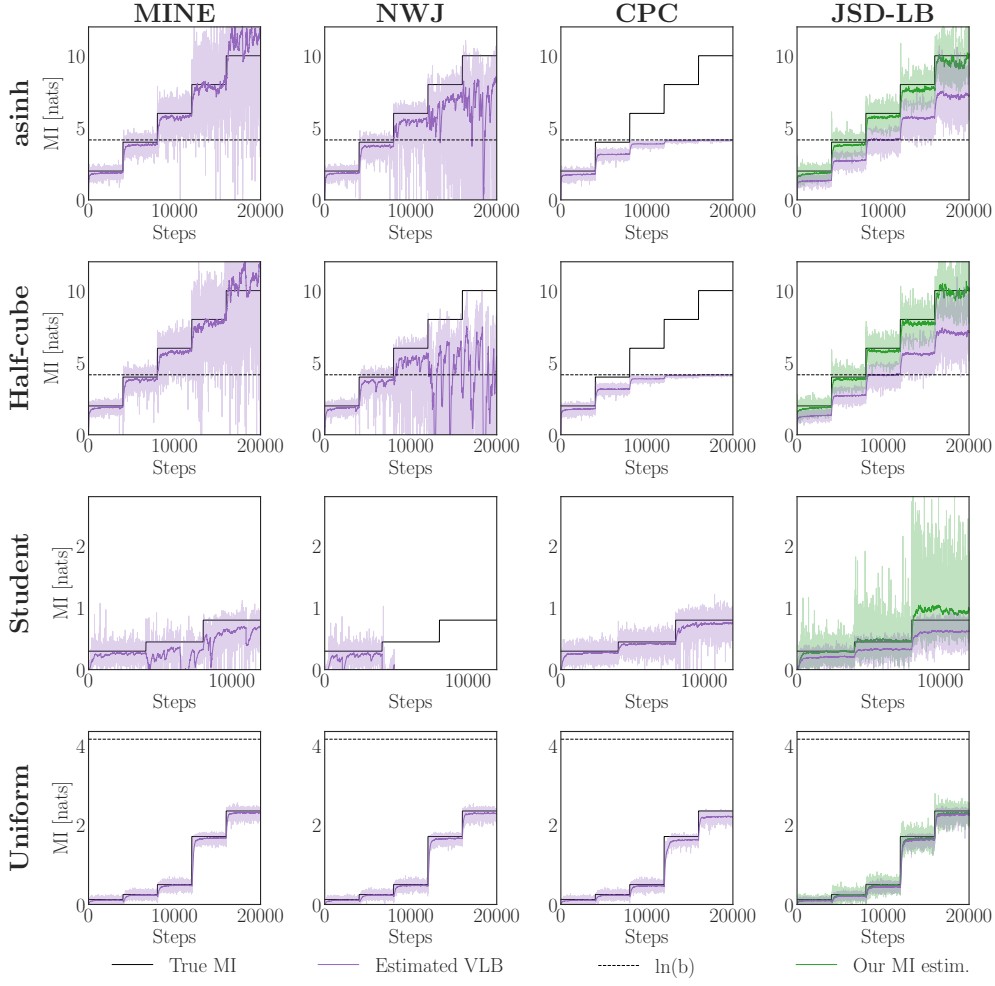

Figure 4: Staircase MI estimation comparison with batch size $b = 64$ for the asinh ($d = 5$), half-cube ($d = 5$), Student ($d = 5$), and uniform distributions.

## 5.2 Complex Gaussian and non-Gaussian distributions

We next evaluate our lower bound in continuous domains where MI is estimated from samples using neural networks. Our goal is to assess how tight the bound remains when used as a variational objective and how well it compares to other neural MI VLB under complex data distributions.

**Setup and metrics** We follow the protocol of Letizia *et al.* [12], who benchmark MI estimators in scenarios where the true MI is known analytically. For each estimator, we report the value of the lower bound objective and the ground truth MI. For our method, we additionally report the estimated MI obtained using the procedure described in Section 4.3 and equivalent to GAN-DIME [12].

All estimators are implemented using a fully connected discriminator with input dimension $2d$, two hidden layers of 256 ReLU units, and one scalar output. Training is performed for 4000 steps using the Adam optimizer and batch size $N = 64$, matching the architecture and hyperparameters from prior work for comparability. As the function $\Xi$ is strictly increasing, maximizing $\Xi \left( \log 2 - \mathcal{L}_{CE} \right)$ is equivalent to minimizing the cross-entropy loss $\mathcal{L}_{CE}$. Therefore, the approximation of $\Xi$ is not used during optimization. Implementation details[2] are available in Appendix E.2.

We compare our approach against other differentiable variational lower bounds that, unlike two-step estimators, can be directly optimized as objective functions for mutual information maximization:

1. **MINE** [3], based on the Donsker-Varadhan dual representation of the KLD;
2. **NWJ** [16], based on another KLD dual representation, which is unbiased but less tight compared to MINE;
3. **CPC** [18], based on contrastive predictive coding with the InfoNCE lower bound;
4. **JSD-LB** (*ours*), based on $I_{CE}$ from (27) with the estimated expected cross-entropy loss.

**Gaussian benchmarks** In the **Gaussian** setting, we define two random variables $U \sim \mathcal{N}(0, I_d)$ and $N \sim \mathcal{N}(0, I_d)$, sampled independently, and construct $V = \rho \, U + \sqrt{1 - \rho^2} \, N$, where $\rho$ is the correlation coefficient. In the **cubic** case, we apply a nonlinear transformation $v \mapsto v^3$ to $V$. This transformation introduces non-Gaussian marginals while keeping the same mutual information. In both cases, the MI is known: $\mathrm{I}\left[U; V\right] = -\frac{d}{2} \log(1 - \rho^2)$.

Figure 3 presents our results. In the **Gaussian** setting, our JSD-LB consistently provides a tight and stable lower bound on the true MI, outperforming MINE, NWJ, and CPC, especially in high-MI regimes. While MINE and NWJ exhibit high variance, JSD-LB and CPC remain more stable. In particular, MINE's instability can lead to significant overestimation of its lower bound, sometimes even exceeding the true MI, whereas CPC remains limited by the contrastive bound $\log(b)$. In contrast, the estimated JSD-LB remains below the true MI, as expected for a valid lower bound, yet closely tracks it. Furthermore, our derived two-step MI estimator from (28), shown in green in Figures 3, 4 and equivalent to GAN-DIME [12], surpasses other variational lower bound estimators. In the **cubic** setting, JSD-LB provides a tight lower bound, while maintaining low-variance estimates compared to MINE and NWJ. Moreover, our two-step MI estimator achieves the most accurate MI estimation, outperforming all other VLBs.

**Complex Gaussian and non-Gaussian benchmarks** We also replicate the more challenging benchmark from Letizia *et al.* [12], following recommendations from Czyż *et al.* [5] using complex Gaussian transformations (**half-cube** and **asinh**, Fig 4) and non-Gaussian distributions (**Student** and **uniform** distributions, Fig 4). Across these settings (see details in Appendix E.2), our lower bound and its derived MI estimations remain competitive, with lower variance VLBs, confirming that the proposed JSD-based objective generalizes beyond Gaussian settings.

## 5.3 Variational Information Bottleneck Experiments

To assess the practical utility of our JSD-LB in representation learning, we evaluate it within the Information Bottleneck (IB) framework [1, 19], which aims to learn compact yet predictive representations $T$ from inputs $U$ for targets $V$, by maximizing $\mathrm{I}\left[T; V\right] - \mathrm{I}\left[T; U\right]$. We reproduce the

---

[2]`https://github.com/ReubenDo/JSDlowerbound`

Table 1: Generalization performance (%) on MNIST dataset. Performance is evaluated by the mean classification accuracy on the MNIST test set after training on the MNIST training set.

| Method | VIB [1] | NIB [9] | squared-VIB [21] | squared-NIB [21] | DisenIB [19] | JSD-LB |
|--------|---------|---------|------------------|------------------|--------------|--------|
| **Testing** | 97.6 | 97.2 | 96.2 | 93.3 | 98.2 | **98.8** |

Table 2: Adversarial robustness performance (%) on MNIST dataset.

| | Method | VIB [1] | NIB [9] | squared-VIB [21] | squared-NIB [21] | DisenIB [19] | JSD-LB |
|--|--------|---------|---------|------------------|------------------|--------------|--------|
| **Training** | $\varepsilon = 0.1$ | 74.1 | 75.2 | 42.1 | 61.3 | 94.3 | **99.5** |
| | $\varepsilon = 0.2$ | 19.1 | 21.8 | 8.7 | 24.1 | 81.5 | **96.0** |
| | $\varepsilon = 0.3$ | 3.5 | 3.2 | 5.9 | 9.3 | 68.4 | **91.4** |
| **Testing** | $\varepsilon = 0.1$ | 73.4 | 75.2 | 42.7 | 62.0 | 90.2 | **94.6** |
| | $\varepsilon = 0.2$ | 20.8 | 23.6 | 9.2 | 24.5 | 80.0 | **89.6** |
| | $\varepsilon = 0.3$ | 4.2 | 3.4 | 5.9 | 9.9 | 67.8 | **86.1** |

Table 3: Distinguishing in- and out-of-distribution test data for MNIST image classification (%). ↑ (resp., ↓) indicates that a larger (resp., lower) value is better.

| Method | VIB [1] | NIB [9] | squared-VIB [21] | squared-NIB [21] | DisenIB [19] | JSD-LB |
|--------|---------|---------|------------------|------------------|--------------|--------|
| TPR95 ↓ | 27.4 | 34.4 | 49.9 | 47.5 | **0.0** | **0.00** |
| AUROC ↑ | 94.6 | 94.2 | 86.6 | 85.6 | 99.4 | **100.0** |
| AUPR In ↑ | 94.8 | 95.2 | 83.5 | 83.3 | 99.6 | **99.9** |
| AUPR Out ↑ | 93.7 | 91.8 | 83.2 | 83.3 | 98.9 | **99.9** |
| Detection Err ↓ | 11.5 | 11.9 | 20.0 | 15.0 | 1.7 | **0.1** |

MNIST experiments of Pan *et al.* [19], where $U$ denotes an image and $V$ its label, using the same architecture and setup. In our setting (JSD-LB), I $[T; V]$ is optimized by maximizing the JSD via the cross-entropy loss. Following Pan *et al.* [19], we assess generalization, adversarial robustness, and out-of-distribution (OOD) detection. Robustness is evaluated by computing the average classification accuracy on adversarially perturbed inputs, where pixel-wise pertubation are obtained using a one-step gradient-based attack with perturbation magnitudes $\epsilon \in \{0.1, 0.2, 0.3\}$. OOD detection is tested on synthetic Gaussian noise samples, using standard OOD detection metrics including True Positive Rate at $95\%$(TPR95), Area Under the ROC Curve (AUROC), Area Under the Precision-Recall Curve (AUPR) and Detection Error.

Our results presented in Tables 1, 2, and 3 show that replacing the MI estimator with JSD-LB leads to state-of-the-art performance in terms of generalization, adversarial robustness, and out-of-distribution robustness. Specifically, it outperforms other Variational Information Bottleneck objectives [1, 9, 21] and the Disentangled Information Bottleneck approach [19]. This set of experiments shows that maximizing JSD via the cross-entropy loss has practical value for representation learning.

## 6  Conclusion

In this work, we addressed a fundamental gap in the theory of representation learning: the lack of formal guarantees linking Jensen–Shannon divergence (JSD), widely used in practice, to Kullback–Leibler divergence (KLD), the basis of MI. We derived a new and optimal lower bound on KLD as a function of JSD, and showed how this leads to a provable lower bound on MI when comparing joint and marginal distributions. Furthermore, we demonstrated that minimizing the cross-entropy loss of a binary classifier distinguishing between joint and marginal pairs increases a lower bound on the MI, unifying existing discriminator-based objectives in representation learning with an information-theoretic interpretation. Through extensive experiments in both discrete and continuous domains, we showed that our bound is tight and also performs competitively as a variational objective and as an estimator in practice. Moreover, we demonstrate its usefulness in representation learning within the Information Bottleneck (IB) framework, reaching state-of-the-art performance. Altogether, this work provides a rigorous theoretical foundation and a practical method for maximizing mutual information through discriminative representation learning.

## Acknowledgment

We thank the anonymous reviewers for their insightful comments and helpful suggestions, which improved the quality of this paper. We would also like to thank Yury Polyanskiy for pointing us to the joint range. R.D. received a Marie Skłodowska-Curie grant No 101154248 (project: SafeREG). This work was performed using HPC resources from GENCI–IDRIS (Grant 2024-SafeREG, 2025-SafeREG). The research leading to these results has received funding from the program "Investissements d'avenir" ANR-10-IAIHU-06. W.W. is supported by the National Institutes of Health (P41EB028741 and R01EB032387).

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

# A   Appendix: Related MI estimators

In this section, we present some existing methods and their formulas for estimating MI. As mentioned in the "Related work" Section 2, these methods can be categorized as differentiable variational lower bounds and two-step (optimization then estimation) methods.

## A.1   Other differentiable Variational Lower Bounds (VLBs)

In this subsection, we review some of the main Variational Lower Bounds (VLBs) on MI. These differentiable bounds can be used directly as minimization criteria in the context of representation learning.

**MINE [3]**   The mutual information neural estimator (MINE) is a VLB on MI derived from the Donsker-Varadhan dual representation of the KLD [20] :

$$\mathrm{I}\left[U; V\right] = \sup_{\theta} \left\{ I_{\mathrm{MINE}}(\theta) \doteq \mathbb{E}_{(u,v)\sim p_{UV}(u,v)}\left[T_{\theta}\left(u,v\right)\right] - \log\left(\mathbb{E}_{(u,v)\sim p_U(u)p_V(v)}\left[e^{T_{\theta}(u,v)}\right]\right) \right\}, \tag{30}$$

where $\theta$ are the parameters of the functions $T_{\theta}: \mathcal{U} \times \mathcal{V} \mapsto \mathbb{R}$ which are defined using a deep neural network. A key limitation of MINE is that it considers the logarithm of the expectation, which leads to biased estimation when approximated using data samples.

**NWJ [16]**   Another VLB based on the KLD dual representation:

$$\mathrm{I}\left[U; V\right] = \sup_{\theta} \left\{ I_{\mathrm{NWJ}}(\theta) \doteq \mathbb{E}_{(u,v)\sim p_{UV}(u,v)}\left[T_{\theta}(u,v)\right] - \mathbb{E}_{(u,v)\sim p_U(u)p_V(v)}\left[e^{T_{\theta}(u,v)-1}\right] \right\} . \tag{31}$$

This VLB is not as tight as MINE, i.e. $I_{\mathrm{NWJ}}(\theta) \leq I_{\mathrm{MINE}}(\theta) \leq \mathrm{I}\left[U; V\right]$ [14], but its estimation is unbiased.

**CPC [18]**   This VLB, based on contrastive predictive coding (CPC), is defined as

$$I_{\mathrm{CPC}}(\theta) = \mathbb{E}_{(u,v)\sim p_{UV;b}(u,v)}\left[ \frac{1}{N}\sum_{i=1}^{b}\log\left( \frac{e^{T_{\theta}(u_i,v_i)}}{\frac{1}{b}\sum_{j=1}^{b}e^{T_{\theta}(u_i,v_j)}} \right) \right] , \tag{32}$$

where $b$ is the batch size and $p_{UV;b}$ denotes the joint distribution of $b$ i.i.d. samples drawn from $p_{UV}$. While CPC estimates have typically low variances, they are upper bounded by $\log b$, which introduces bias and restricts its ability to estimate high mutual information.

## A.2   Two-step MI estimators

Alternatively, another class of estimators has been proposed. Unlike the previously presented VLB, their estimation is two-step. First, a neural network is trained to approximate a function that is later used for estimating the MI. These methods are, therefore, not as easily applied to representation learning, as that would involve solving bilevel (nested) optimization problems.

**SMILE [24]**   Since MINE suffers from high-variance estimations, Song & Ermon [24] proposed to clip the outputs of the neural network $T_{\theta}$, which estimates the log-density ratio, when estimating the partition function:

$$I_{\mathrm{SMILE}}(\theta) \doteq \mathbb{E}_{(u,v)\sim p_{UV}(u,v)}\left[T_{\theta}\left(u,v\right)\right] - \log\left(\mathbb{E}_{(u,v)\sim p_U(u)p_V(v)}\left[\mathrm{clip}\left(e^{T_{\theta}(u,v)}, e^{-\tau}, e^{\tau}\right)\right]\right) \tag{33}$$

where the parameter $\tau \geq 0$ is a hyperparameter and the clip function is defined as follows:

$$\mathrm{clip}(v, l, u) = \max\left(\min\left(v, u\right), l\right) . \tag{34}$$

The SMILE estimate $I_{\mathrm{SMILE}}$ is not a VLB but converges to the $I_{\mathrm{MINE}}$ VLB when $\tau \to \infty$.

In practice, the implementation of SMILE is significantly different from MINE. Indeed, Song & Ermon [24] proposed to train the neural network $T_{\theta}$, by maximizing a Jensen-Shannon MI estimator from [17], as in the Deep InfoMax procedure [8]. SMILE is, therefore, a two-step estimator.

**NJEE [22]**   The Neural Joint Entropy Estimator (NJEE) [22] estimates the MI between discrete random variables by modeling entropy estimation as a sequence of classification tasks. Let $U_m$ denote the $m$-th component of a discrete random vector $U \in \mathbb{R}^d$, and define $U^k$ as the subvector containing the first $k$ components of $U$. Let $V$ be another random variable. Given a batch of $b$ samples, NJEE estimates the entropy of $U$ by training a series of classifiers. Specifically, for each $m$, two neural networks $G_{\theta_m}(U_m \mid U^{m-1})$ and $G_{\theta_m}(U_m \mid V, U^{m-1})$ are trained to predict $U_m$ respectively from its preceding components $U^{m-1}$, and from both $V$ and $U^{m-1}$, using cross-entropy as the loss function. Once trained, the NJEE mutual information estimator is computed as:

$$I_{\text{NJEE}}(U;V) = \hat{H}_b(U_1) + \sum_{m=2}^{d} \text{CE}\left(G_{\theta_m}(U_m \mid U^{m-1})\right) - \sum_{m=1}^{d} \text{CE}\left(G_{\theta_m}(U_m \mid V, U^{m-1})\right), \quad (35)$$

where $\hat{H}_b(U_1)$ is the estimated marginal entropy of the first component. This two-stage procedure is designed for discrete random variables and does not directly extend to the continuous case.

**$f$-DIME [12]**    , is an $f$-divergence based two step method that decouples the optimization of a deep neural network $T_\theta$ and the estimation of the MI. Let $f$ denote a convex lower semicontinuous function $f$ that satisfies $f(1) = 0$. The Fenchel conjugate of $f$ is denoted $f^*$. Specifically, the network $\hat{T}$ is trained to maximize the following quantity:

$$\mathcal{J}_f(T) = \mathbb{E}_{(u,v) \sim p_{UV}(u,v)}\left[T(u,v) - f^*\left(T\left(x, \sigma\left(y\right)\right)\right)\right] \quad (36)$$

and the MI is then estimated as:

$$I_{f\text{-DIME}} = \mathbb{E}_{(u,v) \sim p_{UV}(u,v)}\left[\log\left((f^*)'\left(\hat{T}(u,v)\right)\right)\right]. \quad (37)$$

In practice, Letizia *et al.* [12] proposed to use three $f$ functions that are related to the Kullback–Leibler divergence, GAN, and Hellinger distance squared:

$$f_{KL} : u \rightarrow u\log(u)$$
$$f_{GAN} : u \rightarrow u\log u - (u+1)\log(u+1) + \log 4$$
$$f_{HD} : u \rightarrow \left(\sqrt{u} - 1\right)^2.$$

Experimentally, they found that $f_{GAN}$ achieves the best performance. Importantly, the $f_{GAN}$ is closely related to the $f$ function associated with the Jensen-Shannon Divergence defined as:

$$f_{JSD} : u \rightarrow \frac{1}{2}\left(u\log u - (u+1)\log(\frac{u+1}{2})\right). \quad (38)$$

# B   Appendix: Proofs of theorems and derivations

## B.1   Proof of Theorem 4.1

We begin by deriving the expression for the JSD between two Bernoulli distributions.

**Lemma B.1.** *Let $p = B(\mu)$ and $q = B(\nu)$ be two Bernoulli distributions with parameters $\mu$ and $\nu$. Then, the Jensen-Shannon Divergence between $p$ and $q$ is given by:*

$$\begin{aligned}
D_{\text{JS}}[B(\mu) \mid\mid B(\nu)] = \log 2 &+ \frac{1}{2}\left[\mu\log\frac{\mu}{\mu+\nu} + \nu\log\frac{\nu}{\mu+\nu}\right] \\
&+ \frac{1}{2}\left[(1-\mu)\log\frac{1-\mu}{2-\mu-\nu} + (1-\nu)\log\frac{1-\nu}{2-\mu-\nu}\right]. \quad (39)
\end{aligned}$$

*Proof.* The Kullback-Leibler divergence between two Bernoulli distributions $p$ and $q$ with parameters $\mu$ and $\nu$ is given by:

$$D_{\text{KL}}[p \mid\mid q] = \mu\log\frac{\mu}{\nu} + (1-\mu)\log\frac{1-\mu}{1-\nu}. \quad (40)$$

Given that the mixture $\frac{1}{2}(p+q)$ is a Bernoulli distribution with parameter $m = \frac{p+q}{2}$, the JSD is defined as:

$$\begin{aligned}
D_{JS}[B(\mu) \mid\mid B(\nu)] &= \frac{1}{2}D_{KL}[B(\mu) \mid\mid B(m)] + \frac{1}{2}D_{KL}[B(\nu) \mid\mid B(m)] \\
&= \frac{1}{2}\left[\mu \log \frac{2\mu}{\mu+\nu} + (1-\mu)\log \frac{2(1-\mu)}{2-\mu-\nu}\right] \\
&\quad + \frac{1}{2}\left[\nu \log \frac{2\nu}{\mu+\nu} + (1-\nu)\log \frac{2(1-\nu)}{2-\mu-\nu}\right] \\
&= \log 2 + \frac{1}{2}\left[\mu \log \frac{\mu}{\mu+\nu} + \nu \log \frac{\nu}{\mu+\nu}\right] \\
&\quad + \frac{1}{2}\left[(1-\mu)\log \frac{1-\mu}{2-\mu-\nu} + (1-\nu)\log \frac{1-\nu}{2-\mu-\nu}\right] .
\end{aligned}$$

$\square$

**Lemma B.2.** *The function $\Xi^{-1}$ defined by*

$$\begin{aligned}
\Xi^{-1} : \; &\mathbb{R}_+ \longrightarrow [0, \log 2) \\
&y \longmapsto \log 2 - \tfrac{1}{2}\left[(1+e^{-y})\log(1+e^{-y}) + ye^{-y}\right],
\end{aligned}$$ (41)

*is $C^\infty$, strictly increasing, and satisfies*

$$\Xi^{-1}(0) = 0, \qquad \lim_{y \to +\infty} \Xi^{-1}(y) = \log 2. \tag{42}$$

*Hence, $\Xi^{-1}$ is a bijection from $\mathbb{R}_+$ onto $[0, \log 2)$.*

*Proof.* The function $\Xi^{-1}$ is obtained by finite compositions, additions and multiplication of elementary $C^\infty$ functions. Therefore, $\Xi^{-1} \in C^\infty(\mathbb{R}_+)$.

Differentiating, we obtain

$$(\Xi^{-1})'(y) = -\tfrac{1}{2}\left[-e^{-y}\log(1+e^{-y}) - (1+e^{-y})\tfrac{e^{-y}}{1+e^{-y}} + e^{-y} - ye^{-y}\right] \tag{43}$$

$$= \tfrac{1}{2}e^{-y}\left[\log(1+e^{y})\right]. \tag{44}$$

Since $e^{-y} > 0$ and $\log(1+e^y) > 0$ for all $y > 0$, it follows that $(\Xi^{-1})'(y) > 0$. Hence, $\Xi^{-1}$ is strictly increasing on $\mathbb{R}_+$.

Moreover, as $y \to +\infty$, $e^{-y} \to 0$ and thus $\lim_{y \to +\infty} \Xi^{-1}(y) = \log 2$. At $y = 0$, we have

$$\Xi^{-1}(0) = \log 2 - \tfrac{1}{2}[(1+1)\log(2) + 0] = 0.$$

By continuity and strict monotonicity, the image of $\Xi^{-1}$ is therefore $[0, \log 2)$, establishing bijectivity.

$\square$

**Lemma B.3.** *The function $\Xi$, defined as the inverse of $\Xi^{-1}$ in Eq. (41), is $C^\infty$ and strictly increasing.*

*Proof.* Since $\Xi^{-1}$ is $C^\infty$ with strictly positive derivative on $\mathbb{R}_+$, the inverse function theorem implies that $\Xi$ is $C^\infty$ and strictly increasing on $[0, \log 2)$.

$\square$

We will now prove the main result of this work.

We denote $\Omega$ the lower triangle $\Omega = \{(\mu, \nu) \in [0,1] \times (0,1] \mid \nu \le \mu\}$.

**Lemma B.4.** *Let the mapping $\phi$ be defined as on the lower triangle:*

$$\phi : (\mu, \nu) \in \Omega \mapsto (j(\mu, \nu), k(\mu, \nu)) \tag{45}$$

*where $j$ and $k$ are defined as follows:*

$$j(\mu, \nu) \doteq D_{JS}[B(\mu) \mid\mid B(\nu)] \quad and \quad k(\mu, \nu) \doteq D_{KL}[B(\mu) \mid\mid B(\nu)] . \tag{46}$$

*Then, the Jacobian $J$ of the mapping $\phi$ is given by:*

$$J(\mu, \nu) = \begin{bmatrix} \frac{\partial j}{\partial \mu} & \frac{\partial j}{\partial \nu} \\ \frac{\partial k}{\partial \mu} & \frac{\partial k}{\partial \nu} \end{bmatrix}(\mu, \nu) = \begin{bmatrix} \frac{1}{2}\left[\mathbb{L}(\mu) - \mathbb{L}(m)\right] & \frac{1}{2}\left[\mathbb{L}(\nu) - \mathbb{L}(m)\right] \\ \mathbb{L}(\mu) - \mathbb{L}(\nu) & -\left[\frac{\mu}{\nu} - \frac{1-\mu}{1-\nu}\right] \end{bmatrix} \tag{47}$$

*where $\mathbb{L}$ is the logit function.*

*Proof.* Using Lemma B.1, the partial derivative $\frac{\partial j}{\partial \mu}$ is given by:

$$\frac{\partial j}{\partial \mu}(\mu, \nu) = \frac{1}{2}\left[\mu\left(\frac{1}{\mu} - \frac{1}{\mu+\nu}\right) + \log\frac{\mu}{\mu+\nu} - \frac{\nu}{\mu+\nu}\right]$$

$$+ \frac{1}{2}\left[(1-\mu)\left(\frac{1}{2-\mu-\nu} - \frac{1}{1-\mu}\right) - \log\frac{1-\mu}{2-\mu-\nu} + \frac{1-\nu}{2-\mu-\nu}\right]$$

$$= \frac{1}{2}\left[\log\frac{\mu}{\mu+\nu} - \log\frac{1-\mu}{2-\mu-\nu} + 1 - \frac{\mu}{\mu+\nu} - \frac{\nu}{\mu+\nu} + \frac{1-\mu}{2-\mu-\nu} - 1 + \frac{1-\nu}{2-\mu+\nu}\right]$$

$$= \frac{1}{2}\left[\log\frac{\mu}{\mu+\nu} - \log\frac{1-\mu}{2-\mu-\nu}\right]$$

$$= \frac{1}{2}\left[\log\frac{\mu}{1-\mu} - \log\frac{m}{1-m}\right]$$

$$= \frac{1}{2}\left[\mathbb{L}(\mu) - \mathbb{L}(m)\right] .$$

By symmetry of the JSD:

$$\frac{\partial j}{\partial \nu}(\mu, \nu) = \frac{1}{2}\left[\mathbb{L}(\nu) - \mathbb{L}(m)\right] . \tag{48}$$

Finally, using Eq (40):

$$\frac{\partial k}{\partial \mu}(\mu, \nu) = \log\frac{\mu(1-\nu)}{\nu(1-\mu)} \tag{49}$$

$$= \mathbb{L}(\mu) - \mathbb{L}(\nu) , \tag{50}$$

and

$$\frac{\partial k}{\partial \nu}(\mu, \nu) = -\left[\frac{\mu}{\nu} - \frac{1-\mu}{1-\nu}\right] . \tag{51}$$

$\square$

**Conjecture B.5.** *The determinant of the Jacobian $J$ defined in Eq.(47) is negative on the interior $\Omega^{\mathrm{o}} = \{(\mu, \nu) \in (0,1) \times (0,1) \mid \nu < \mu\}$ of $\Omega$, i.e.:*

$$\forall (\mu, \nu) \in \Omega^{\mathrm{o}}, \quad \det(J(\mu, \nu)) < 0 . \tag{52}$$

While we have not found an analytical proof of the conjecture, it can be verified numerically, as shown in Figure 5. The negative of the determinant of the Jacobian is always positive.

**Theorem B.6.** *The optimal lower bound on Kullback-Leibler divergence established by the Jensen-Shannon divergence is:*

$$\Xi\left(\mathrm{D}_{\mathrm{JS}}[p \,||\, q]\right) \leq \mathrm{D}_{\mathrm{KL}}[p \,||\, q] \tag{53}$$

*where $\Xi : [0, \log 2) \mapsto \mathbb{R}^+$ is a strictly increasing function defined implicitly by its inverse,*

$$\Xi^{-1} : y \mapsto \mathrm{D}_{\mathrm{JS}}[B(1) \,||\, B\left(\exp\left(-y\right)\right)], \tag{54}$$

*where $B\left(\exp\left(-y\right)\right)$ is a Bernoulli distribution with parameter $\exp\left(-y\right) \in (1/2, 1]$. $\Xi^{-1}$ has a closed-form expression related to Eq. (17).*

*Proof.* Following the Theorem 3.1 by Harremoës & Vajda [7], the joint range between the JSD and the KLD can be fully characterized by restricting $p$, $q$ to be pairs of Bernoulli distributions compatible with JSD and KLD,

$$x = \mathrm{D}_{\mathrm{JS}}[B(\mu) \,||\, B(\nu)] \quad \text{and} \quad y = \mathrm{D}_{\mathrm{KL}}[B(\mu) \,||\, B(\nu)] , \tag{55}$$

for $(\mu, \nu) \in [0, 1] \times (0, 1) \cup \{(0, 0), (1, 1)\}$, where $B(\theta)$ is a Bernoulli distribution with parameter $\theta$.

We are therefore interested in the image of the mapping $\phi$ defined as:

$$\phi : (\mu, \nu) \mapsto (\mathrm{D}_{\mathrm{JS}}[B(\mu) \,||\, B(\nu)], \mathrm{D}_{\mathrm{KL}}[B(\mu) \,||\, B(\nu)]) \tag{56}$$

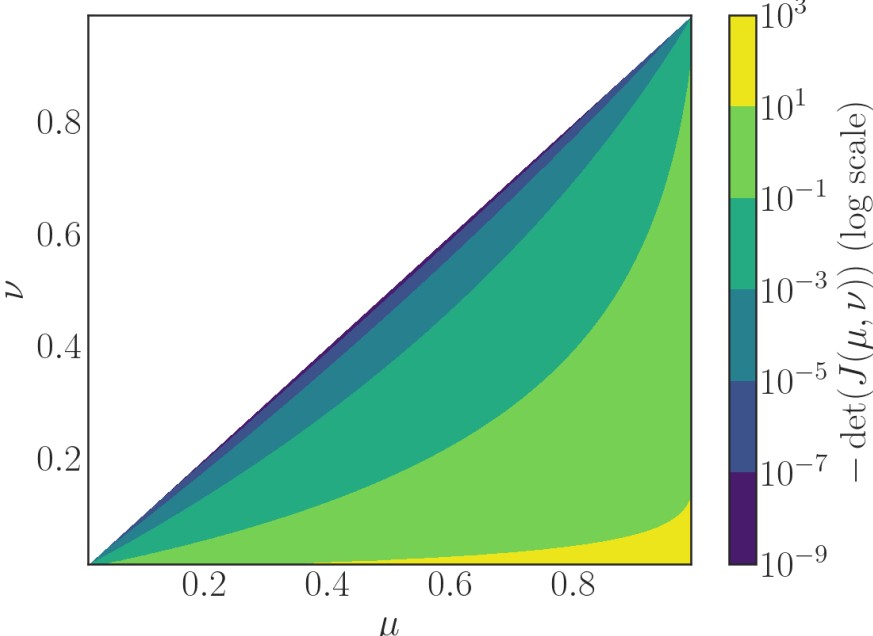

Figure 5: Visualization of the negative of the determinant of the Jacobian. We can observe that it is constantly positive, meaning that the determinant of the Jacobian is always negative.

from the unit square $[0, 1] \times (0, 1) \cup \{(0, 0), (1, 1)\}$ into $\mathbb{R}^2$.

Exploiting the symmetry of the divergences:

$$D_{\mathrm{KL}}[B(\mu) \,||\, B(\nu)] = D_{\mathrm{KL}}[B(1 - \mu) \,||\, B(1 - \nu)], \tag{57}$$
$$D_{\mathrm{JS}}[B(\mu) \,||\, B(\nu)] = D_{\mathrm{JS}}[B(1 - \mu) \,||\, B(1 - \nu)], \tag{58}$$

we restrict our analysis to the image $\phi(\Omega)$ of the lower triangle $\Omega = \{(\mu, \nu) \in [0, 1] \times (0, 1] \,|\, \nu \leq \mu\}$.

Using Conjecture B.5, the determinant of the Jacobian is negative for all inner points in $\Omega$. According to the open mapping theorem, the inner points are therefore mapped into interior points of the range. To determine the lower envelope of the image, we examine the image of the boundary $\delta\Omega = (S_1, S_2, S_3)$. Given that, $S_1$ and $S_2$ respectively map to a single point $(0, 0)$ and a horizontal line $y = \infty$, $S_3$ defines the lower envelope of the range and is, therefore, our bound.

## B.2 Proof of Eq. (26)

The variational representation [17, 27] of can be expressed as:

$$I_{\mathrm{JS}}[U; V] = D_{\mathrm{JS}}[p_{UV} \,||\, p_U \otimes p_V] = \tfrac{1}{2} \max_{t} \left[ \mathbb{E}_{p_{UV}}[t] - \mathbb{E}_{p_U \otimes p_V}[-\log(2 - \exp(t))] \right] \tag{59}$$

where $t : \mathcal{U} \times \mathcal{V} \to (-\infty, \log 2)$ is a variational function.

Now, define the following reparameterization,

$$t(u, v) \doteq \log(2 q_\theta(z = 1 \,|\, u, v)), \tag{60}$$

where $q_\theta(z = 1 \,|\, u, v)$ is a neural network.

Substituting (60) into (59), we obtain (25):

$$I_{JS}[U;V] = \frac{1}{2} \max_t \left[ \mathop{\mathbb{E}}_{p_{UV}} \left[ \log(2q_\theta(z=1 \mid u,v)) \right] - \mathop{\mathbb{E}}_{p_U \otimes p_V} \left[ -\log(2 - 2q_\theta(z=1 \mid u,v)) \right] \right] \quad (61)$$

$$= \frac{1}{2} \max_\theta \left[ \log 4 + \mathop{\mathbb{E}}_{p_{UV}} \left[ \log q_\theta(z=1|u,v) \right] + \mathop{\mathbb{E}}_{p_U \otimes p_V} \left[ \log(1 - q_\theta(z=1|u,v)) \right] \right] \quad (62)$$

$$(63)$$

The expected cross-entropy (CE) under our classifier model in Eq. (21) is defined as:

$$\mathcal{L}_{CE} \doteq \mathop{\mathbb{E}}_{(u,v,z) \sim p_{UVZ}} \left[ -\log q_\theta(z|u,v) \right] = \mathop{\mathbb{E}}_{z \sim p_Z} \left[ \mathop{\mathbb{E}}_{(u,v) \sim p_{UV|Z}} \left[ -\log q_\theta(z|u,v) \right] \right] \quad (64)$$

or, expanding the expectation over $z$,

$$\mathcal{L}_{CE} = \frac{1}{2} \mathop{\mathbb{E}}_{(u,v) \sim p_{UV}} \left[ -\log q_\theta(z=1|u,v) \right] + \frac{1}{2} \mathop{\mathbb{E}}_{(u,v) \sim p_U \otimes p_V} \left[ -\log q_\theta(z=0|u,v) \right] . \quad (65)$$

With Eqn. (62) we obtain

$$I_{JS}[U;V] \geq \log 2 - \min_\theta \mathcal{L}_{CE}(\theta) . \quad (66)$$

$$\square$$

## B.3 Alternative to Section 4.2 to obtain (26) and gap quantification

Here, we revisit the relationship between Jensen-Shannon divergence and the cross-entropy training loss. In our main manuscript, the relationship was derived using properties of $f$-divergences. The following derivation is more discriminator-oriented, and does not use $f$-divergence properties directly.

*Alternative proof to* (26). Recall the classifier model of Equation (21). We use a standard binary classification setup to estimate the JSD between the joint distribution $p_{UV}$ and the product of its marginals $p_U \otimes p_V$. Consider the mixture model:

$$(U,V) \mid Z = 1 \sim p_{UV}(u,v), \quad (U,V) \mid Z = 0 \sim p_U \otimes p_V, \quad \text{with } Z \sim B(1/2). \quad (67)$$

This defines a joint distribution over $(u,v,z)$, denoted $\tilde{p}(u,v,z)$, with marginal over $(u,v)$:

$$\tilde{p}(u,v) = m(u,v) \doteq \tfrac{1}{2}p_{UV}(u,v) + \tfrac{1}{2}p_U(u)p_V(v). \quad (68)$$

Next we study the MI among $(U,V)$ and $Z$ under that model,

$$I[(U,V);Z] \doteq D_{KL}[\tilde{p}_{UVZ} \| p_{UV} \otimes p_Z] \quad (69)$$

$$= \mathop{\mathbb{E}}_{(u,v,z) \sim \tilde{p}_{UVZ}} \left[ \log \frac{\tilde{p}_{UVZ}(u,v,z)}{\tilde{p}_{UV}(u,v)p_Z(z)} \right] \quad (70)$$

$$= \mathop{\mathbb{E}}_{z \sim p_Z} \left[ \mathop{\mathbb{E}}_{(u,v) \sim \tilde{p}_{UV|Z}} \left[ \log \frac{\tilde{p}_{UV|Z}}{m(u,v)} \right] \right] \quad (71)$$

$$(\text{expanding the expectation over } z) \quad (72)$$

$$= \frac{1}{2} \mathop{\mathbb{E}}_{(u,v) \sim p_{UV}} \left[ \log \frac{p_{UV}(u,v)}{m(u,v)} \right] + \frac{1}{2} \mathop{\mathbb{E}}_{(u,v) \sim p_U \otimes p_V} \left[ \log \frac{p_U \otimes p_V(u,v)}{m(u,v)} \right] . \quad (73)$$

Therefore:

$$I[(U,V);Z] = D_{JS}[p_{UV} \| p_U \otimes p_V] = I_{JS}[U;V] . \quad (74)$$

From this, we observe that, in the classifier setup above, the MI between $(U,V)$ and $Z$ equals the Jensen-Shannon divergence between $U$ and $V$. A similar property holds for general JS divergences.

MI also characterizes the information loss by conditioning. In this setup,

$$I[(U,V);Z] = H[Z] - H[Z|U,V] = \log 2 - H[Z|U,V] , \quad (75)$$

where the conditional entropy is defined as

$$\mathrm{H}\left[Z|U,V\right] \doteq \mathop{\mathbb{E}}_{(u,v,z)\sim\tilde{p}_{UVZ}} \left[-\log\tilde{p}_{Z|UV}(z|u,v)\right] \ . \tag{76}$$

Combining Equations (74) and (75) yields

$$\mathrm{I}_{\mathrm{JS}}\left[U;V\right] = \log 2 - \mathrm{H}\left[Z|U,V\right] \ . \tag{77}$$

Suppose we define a variational distribution $q_\theta(z|u,v)$ ; then

$$\mathrm{H}\left[Z|U,V\right] = \mathop{\mathbb{E}}_{(u,v,z)\sim\tilde{p}_{UVZ}} \left[-\log\tilde{p}_{Z|UV}(z|u,v) + \log q_\theta(z|u,v)\right]$$
$$+ \mathop{\mathbb{E}}_{(u,v,z)\sim\tilde{p}_{UVZ}} \left[-\log q_\theta(z|u,v)\right] \tag{78}$$

$$= \mathop{\mathbb{E}}_{(u,v)\sim\tilde{p}_{UV}} \left[\mathop{\mathbb{E}}_{z\sim\tilde{p}_{Z|UV}} \left[-\log\tilde{p}_{Z|UV}(u,v|z) + \log q_\theta(z|u,v)\right]\right]$$
$$+ \mathop{\mathbb{E}}_{(u,v,z)\sim\tilde{p}_{UVZ}} \left[-\log q_\theta(z|u,v)\right] \ . \tag{79}$$

Therefore,

$$\mathrm{H}\left[Z|UV\right] = -\underbrace{\mathop{\mathbb{E}}_{(u,v)\sim\tilde{p}_{UV}} \left[\mathrm{D}_{\mathrm{KL}}[\tilde{p}_{Z|UV}(u,v) \ || \ q_\theta(z|u,v)]\right]}_{\doteq\delta} + \underbrace{\mathop{\mathbb{E}}_{(u,v,z)\sim\tilde{p}_{UVZ}} \left[-\log q_\theta(z|u,v)\right]}_{=\mathcal{L}_{CE}} \ . \tag{80}$$

The first term $\delta$ on the right-hand side of (80) is the expected KLD between the true posterior $p_{Z|UV}$ and the model posterior $q_\theta$, which is always non-negative. The second term corresponds to the expected cross-entropy loss of the classifier $q_\theta$.

Using the non-negativity of the KL divergence, we obtain the inequality:

$$\mathrm{H}\left[Z|UV\right] \geq \mathcal{L}_{CE}, \tag{81}$$

where $\mathcal{L}_{CE}$ denotes the expected cross-entropy loss. Equality holds if and only if the model distribution $q_\theta(z \mid u,v)$ is equal to the true posterior $p_{Z|UV}(z \mid u,v)$ almost everywhere.

Combining this with Equation (77) yields

$$\mathrm{I}_{\mathrm{JS}}\left[U;V\right] \geq \log 2 - \mathcal{L}_{CE} \ . \tag{82}$$

$\square$

Thanks to this new derivation, we can explicitly quantify the gap between the true JSD lower bound $\Xi(\mathrm{I}_{\mathrm{JS}}\left[U;V\right])$ and the one obtained from the cross-entropy $I_{CE}(\theta)$ as follows:

**Lemma B.7.** *Let $q_\theta$ denote the discriminator, and let $\mathcal{L}_{CE}(\theta)$ be its expected binary cross-entropy. The gap between our new variational lower bound $I_{CE}(\theta)$ and the optimal lower bound $\Xi\left(\mathrm{I}_{\mathrm{JS}}\left[U;V\right]\right)$ is given by*

$$\Xi\left(\mathrm{I}_{\mathrm{JS}}\left[U;V\right]\right) - I_{CE}(\theta) \approx \delta \cdot \Xi'\left(I_{CE}(\theta)\right), \tag{83}$$

*where $\delta$ denotes the expected Kullback–Leibler divergence between the true and approximate posteriors:*

$$\mathop{\mathbb{E}}_{(u,v)\sim\tilde{p}_{UV}} \left[\mathrm{D}_{\mathrm{KL}}[\tilde{p}_{Z|UV}(u,v) \ || \ q_\theta(z \mid u,v)]\right] \ . \tag{84}$$

*Proof.* Combining Equations (77) and (80), we obtain

$$\mathrm{I}_{\mathrm{JS}}\left[U;V\right] = \log(2) - \mathcal{L}_{CE}(\theta) + \delta \ . \tag{85}$$

Hence, the gap $\Delta_\theta$ between the optimal JSD lower bound and our bound $I_{CE}(\theta)$ is

$$\Delta_\theta = \Xi\left(\mathrm{I}_{\mathrm{JS}}\left[U;V\right]\right) - \Xi\left(I_{CE}(\theta)\right) \tag{86}$$
$$= \Xi\left(\log(2) - \mathcal{L}_{CE}(\theta) + \delta\right) - \Xi\left(\log(2) - \mathcal{L}_{CE}(\theta)\right) \ . \tag{87}$$

Applying a first-order Taylor expansion around $I_{CE}$ leads to:

$$\Delta_\theta \approx \delta \cdot \Xi'\left(I_{CE}(\theta)\right) \ . \tag{88}$$

$\square$

## B.4 Proof of Eq. (28)

We derive here a method for approximating the MI by way of the trained classifier, building on the setup of Section 4.2 and B.3.

Under with the joint model of Eq. (68) (originally (21)),

$$\tilde{p}_{UV|Z}(u,v|z) = \frac{\tilde{p}_{ZUV}(z,u,v)}{p_Z(z)} \quad \text{and} \quad \tilde{p}_{Z|UV}(z|u,v) = \frac{\tilde{p}_{ZUV}(z,u,v)}{\tilde{p}_{UV}(u,v)} . \tag{89}$$

Moreover, since $Z \sim B(1/2)$, we have $p(z=1) = p(z=0) = \frac{1}{2}$. Therefore,

$$\log \frac{\tilde{p}_{Z|UV}(z=1|u,v)}{\tilde{p}_{Z|UV}(z=0|u,v)} = \log \frac{\tilde{p}_{ZUV}(z=1,u,v)}{\tilde{p}_{ZUV}(z=0,u,v)} = \log \frac{\tilde{p}_{UV|Z}(u,v|z=1)}{\tilde{p}_{UV|Z}(u,v|z=0)} = \log \frac{p_{UV}(u,v)}{(p_U \otimes p_V)(u,v)} . \tag{90}$$

Using the definition of KLD,

$$D_{KL}[p_{UV} \mid\mid p_U \otimes p_V] \doteq \underset{(u,v)\sim p_{UV}}{\mathbb{E}} \left[ \log \frac{p_{UV}(u,v)}{(p_U \otimes p_V)(u,v)} \right] = \underset{(u,v)\sim p_{UV}}{\mathbb{E}} \left[ \log \frac{\tilde{p}_{Z|UV}(z=1|u,v)}{\tilde{p}_{Z|UV}(z=0|u,v)} \right] , \tag{91}$$

and finally,

$$I[U;V] \doteq D_{KL}[p_{UV} \mid\mid p_U \otimes p_V] = \underset{(u,v)\sim p_{UV}}{\mathbb{E}} \left[ \mathbb{L}(\tilde{p}_{Z|UV}(z=1|u,v)] \tag{92}$$

where $\mathbb{L}$ is the logit transform. Thus we can estimate the MI of $p_{UV}$ by first training a classifier to approximate the true posterior, $p_{Z|UV}$, and after that use it to approximate the expectation by averaging over the 'true' data.

## C   Appendix: Approximation of the function $\Xi$

In this section, we derive Equation (17), which is used to define the inverse $\Xi^{-1}$. This allows the use of numerical solvers to approximate $\Xi$. Finally, we present the approximation of the function $\Xi$ using the Logit transform.

### C.1   Derivation of Equation (17)

Let's recall (17):

$$D_{JS}[B(1) \mid\mid B(\nu)] = \log 2 - \frac{1}{2} \left[ \log(1+\nu) - \nu \log \left( \frac{\nu}{1+\nu} \right) \right] \tag{93}$$

*Proof.* Using Lemma (B.1):

$$D_{JS}[B(1) \mid\mid B(\nu)] = \log 2 + \frac{1}{2} \left[ \log \frac{1}{1+\nu} + \nu \frac{\nu}{1+\nu} + (1-\nu) \log \frac{1-\nu}{1-\nu} \right] \tag{94}$$

$$= \log 2 - \frac{1}{2} \left[ \log(1+\nu) - \nu \log \left( \frac{\nu}{1+\nu} \right) \right] \tag{95}$$

$\square$

## C.2 Python code for implementing $\Xi$

```python
from scipy.optimize import root_scalar
import numpy as np

# Define the inverse function Xi(x)
def inv_xi(y):
    z = np.exp(-y)
    term1 = np.log(1 + z)
    term2 = z * np.log(z / (1 + z))
    return np.log(2) - 0.5 * (term1 - term2)

# Define the function Xi(x)
def xi(x):
    result = root_scalar(lambda y: inv_xi(y) - x, bracket=[0, 100],
        method='brentq')
    return result.root if result.converged else None
```

Algorithm 1: Algorithmic implementation of the $\Xi$ function as the inverse of its known inverse $\Xi^{-1}$

## C.3 Proposed differentiable approximation of $\Xi$

To enable the use of our bound within a differentiable objective, we introduce a smooth, differentiable approximation of the $\Xi$ function. This approximation is derived from the Taylor expansion of the KL divergence in terms of the JSD. The lower bound is characterized by the following parametric curve:

$$x = D_{JS}[B(1) \parallel B(\nu)] = \Xi^{-1}(-\log \nu) \tag{96}$$

$$y = D_{KL}[B(1) \parallel B(\nu)] = -\log \nu , \tag{97}$$

where $\Xi : [0, \log 2) \mapsto \mathbb{R}^+$ is a strictly increasing function implicitly defined by its inverse:

$$\Xi^{-1} : y \mapsto \log 2 - \frac{1}{2} \left[ \left(1 + e^{-y}\right) \log \left(1 + e^{-y}\right) + ye^{-y} \right] . \tag{98}$$

Hence, we can express:

$$x = \Xi^{-1}(y) . \tag{99}$$

Let $\mathbb{L} : x \mapsto \log \frac{x}{1-x}$ denote the Logit function. We then observe that:

$$\mathbb{L}\left(\frac{1}{2}\left(\frac{x}{\log 2} + 1\right)\right) = \log \left(\frac{\frac{x}{\log 2} + 1}{1 - \frac{x}{\log 2}}\right) \tag{100}$$

$$= \log \left(\frac{\frac{\Xi^{-1}(y)}{\log 2} + 1}{1 - \frac{\Xi^{-1}(y)}{\log 2}}\right) \tag{101}$$

$$= \log \left(\frac{1 - \frac{1}{2\log 2}\left[(1 + e^{-y})\log(1 + e^{-y}) + ye^{-y}\right]}{\frac{1}{2\log 2}\left[(1 + e^{-y})\log(1 + e^{-y}) + ye^{-y}\right]}\right) \tag{102}$$

$$= \log \left(\frac{4\log 2}{(1 + e^{-y})\log(1 + e^{-y}) + ye^{-y}} - 1\right) \tag{103}$$

The first-order Taylor expansion of the right-hand side of Eq. (103) around $y = 0$ is simply $y \mapsto y$. This motivates approximating $\Xi$ using a scaled logit function. Through a grid search, we found that scaling the logit by a factor of 1.15 yields the lowest median approximation error over $x \in (0, 1)$:

$$\Xi(x) \approx 1.15 \cdot \mathbb{L}\left(\frac{1}{2}\left(\frac{x}{\log 2} + 1\right)\right) . \tag{104}$$

As shown in Figure 6, our proposed approximation closely matches the numerically estimated inverse obtained via a solver. Importantly, this formulation is fully differentiable and can be directly integrated into a variational lower bound (VLB) objective in representation learning.

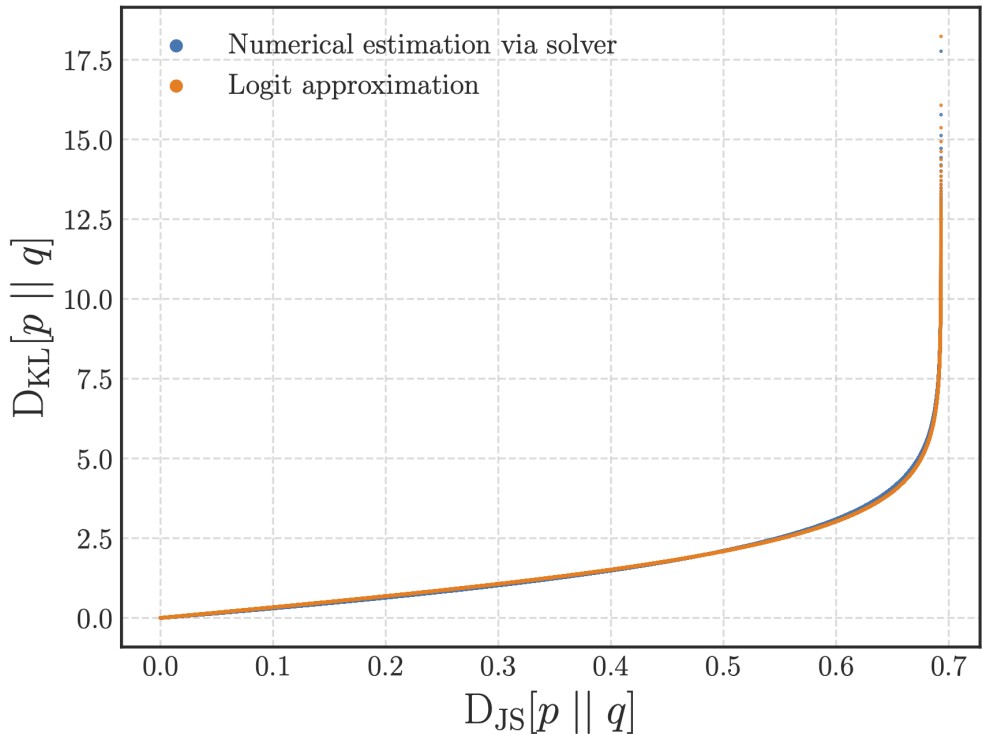

Figure 6: Comparison between the numerical evaluation of $\Xi$ using Algorithm 1 and the proposed differentiable approximation from Eq. (104).

## D   Appendix: Tightness Analysis in the Mutual Information Setting

While our JSD–KLD lower bound in Theorem 4.1 is the tightest possible (i.e., equality is achieved for a family of distributions) between these two divergences for arbitrary distributions, one might expect a tighter bound when restricted to the mutual information setting. i.e., when comparing a joint distribution to the product of its marginals. Remarkably, as illustrated in Figure 2 of the main text, discrete joint–marginal pairs can lie close to our general bound. In this section, we analytically show that there exists an infinite family of distributions in the mutual information setting that exactly lie on our lower bound.

**Lemma D.1.** *Let $(U, V)$ be a pair of uniform categorical variables of dimension $k \in \mathbb{N}^+$ that are fully dependent (i.e., $U = V$). Then:*

$$\mathrm{I}\left[U; V\right] = \Xi(\mathrm{I}_{\mathrm{JS}}\left[U; V\right]) . \tag{105}$$

*Proof.* Let $k \in \mathbb{N}^+$, and assume $U$ and $V$ are fully dependent (i.e., $U = V$) uniform categorical variables of dimension $k$. Their joint and marginal distributions are given by:

$$P_U = P_V = \tfrac{1}{k}\mathbf{1}_k , \quad P_{UV} = \tfrac{1}{k}I_k ,$$

and consequently,

$$P_U \otimes P_V = \tfrac{1}{k^2}\mathbf{1}_{(k,k)} .$$

Let $M = \frac{1}{2}\left(P_U \otimes P_V + P_{UV}\right)$ denote the mixture distribution. Then $M_{i,j} = \frac{1/k + 1/k^2}{2}$ if $i = j$, and $M_{i,j} = \frac{1}{2k^2}$ otherwise.

We now compute $\mathrm{I}\left[U; V\right]$ and $\mathrm{I}_{\mathrm{JS}}\left[U; V\right]$:

$$\mathrm{I}\left[U; V\right] = \sum_{i=1}^{k} \tfrac{1}{k} \log \frac{1/k}{1/k^2} = \log k . \tag{106}$$

$$I_{\text{JS}}[U;V] = \frac{1}{2}\left( \sum_{i=1}^{k} \frac{1}{k} \log \frac{1/k}{\frac{1}{2}(1/k + 1/k^2)} + \sum_{i=1}^{k} \frac{1}{k^2} \log \frac{1/k^2}{\frac{1}{2}(1/k + 1/k^2)} + \sum_{i=1}^{k} \sum_{\substack{j=1 \\ j \neq i}}^{k} \frac{1}{k^2} \log \frac{1/k^2}{1/(2k^2)} \right)$$

(107)

$$= \frac{1}{2}\left( \log \frac{2}{1+1/k} + \frac{1}{k} \log \frac{2/k}{1+1/k} + \frac{k-1}{k} \log 2 \right) \tag{108}$$

$$= \log 2 - \frac{1}{2}\left[ \log(1 + 1/k) - \frac{1}{k} \log \frac{1/k}{1+1/k} \right] \tag{109}$$

$$= \Xi^{-1}(\log k). \tag{110}$$

Combining Equations (106) and (110), we obtain

$$I_{\text{JS}}[U;V] = \Xi^{-1}(I[U;V]), \tag{111}$$

which completes the proof. $\qquad\square$

## E    Appendix: Experimental details

### E.1    Discrete case experiments

In this section, we present a synthetic experiment designed to empirically validate the tightness of our proposed lower bound between mutual information (MI) and the Jensen-Shannon divergence (JSD), as introduced in Theorem 4.1. Unlike general divergences between arbitrary distributions, here we specialize the analysis to the mutual information setting, which corresponds to comparing a joint distribution to the product of its marginals.

We consider a family of discrete joint distributions $P_{UV}^{(\alpha)} \in \mathbb{R}^{k \times k}$ parameterized by a dependence factor $\alpha \in [0,1]$, which interpolates between the independent and perfectly dependent cases. Specifically, we fix the marginals to be uniform categorical distributions: $P_U = P_V = \frac{1}{k}\mathbf{1}_k$, where $k$ denotes the number of categories. The joint distribution is constructed as a convex combination of the independent and deterministic cases:

$$P_{UV}^{(\alpha)} = (1 - \alpha)P_U \otimes P_V + \alpha \cdot \text{diag}(P_U), \tag{112}$$

where $\text{diag}(P_U)$ denotes the diagonal matrix with $P_U$ on its diagonal, ensuring perfect dependence along the diagonal. This construction ensures that:

- When $\alpha = 0$, $U$ and $V$ are independent.
- When $\alpha = 1$, $U = V$ with probability one.

For each $\alpha$, we compute the mutual information $I(U;V) = D_{\text{KL}}[P_{UV}^{(\alpha)} \parallel P_U \otimes P_V]$ and the Jensen-Shannon divergence $D_{\text{JS}}[P_{UV}^{(\alpha)} \parallel P_U \otimes P_V]$, and compare them to the lower bound $\Xi(D_{\text{JS}}[P_{UV}^{(\alpha)} \parallel P_U \otimes P_V])$, where $\Xi$ is defined in Equation (19).

### E.2    Complex Gaussian and non-Gaussian distributions experiments

In this section, we provide further details on the experimental setup described in Section 5.2 and present supplementary results for the linear and cubic Gaussian experiments.

#### E.2.1    Experimental setup

The neural network architectures employed in our experiments follow the joint architecture, as in [12]. We do not use the deranged architecture from [12], as it is incompatible with the CPC objective.

**Joint architecture.** Following the architecture described in [12], we employ a fully connected feed-forward neural network where the input layer has dimensionality $2d$, corresponding to the concatenation of two $d$-dimensional samples. The network comprises two hidden layers, each with 256 neurons and ReLU activations, and a single scalar output neuron. During training, the network

receives as input $b^2$ pairs $(u, v)$ per iteration, where these pairs are formed by taking all combinations of samples $u$ and $v$ independently drawn from the joint distribution $p_{(u,v)}$.

**Staircase** In the Gaussian setting, we define two random variables $U \sim \mathcal{N}(0, I_d)$ and $N \sim \mathcal{N}(0, I_d)$, sampled independently, and construct $V = \rho U + \sqrt{1 - \rho^2} N$, where $\rho$ is the correlation coefficient.

To explore settings with non-Gaussian marginals while preserving the same mutual information, we apply nonlinear transformations to $V$. In the cubic case, we use the mapping $v \mapsto v^3$, which preserves the dependency structure but introduces non-Gaussian marginals. Additionally, we benchmark two alternative transformations: the inverse hyperbolic sine (asinh), which shortens the tails of the distribution, and the half-cube transformation, which stretches them. These mappings allow us to evaluate the robustness of our method under controlled deviations from Gaussianity.

During the training procedure, every 4k iterations, the target value of the MI is increased by 2 nats, for 5 times, obtaining a target staircase with 5 steps. The change in target MI is obtained by increasing $\rho$, which affects the true MI according to $\mathrm{I}\left[U; V\right] = -\frac{d}{2} \log(1 - \rho^2)$.

**Training procedure.** Each neural estimator is trained using Adam optimizer, with learning rate 0.002, $\beta_1 = 0.9$, $\beta_2 = 0.999$. The batch size is initially set to $b = 64$.

### E.2.2 Analysis for different values of batch size $b$

Our VLB is robust to changes in batch size $b$. In Figures 7, 8, and 9, we observe that MINE and NWJ exhibit high variance, even for large batch sizes (e.g., $b = 128$), while JSD-LB and CPC remain more stable. Notably, MINE's instability can cause substantial overestimation of its lower bound, occasionally exceeding the true MI. CPC, on the other hand, provides low-variance estimates but is upper bounded by the contrastive limit $\log(b)$, which results in higher bias in high-MI regimes.

### E.2.3 Analysis for different values of dimension $d$

We additionally perform an analysis for different values of dimension $d$. We report the achieved bias, variance, and mean squared error (MSE) corresponding to the four settings $d \in \{5, 10, 20, 100\}$ in Table 4. We experimentally found that MINE produces estimate with high variance, NWJ is unstable in high dimension, and CPC is bounded by $\ln(b)$ When repurposed for two-step MI estimation, our method surpasses other variational lower bound estimators across dimensions. In contrast, while the bias of JSD-LB increases with dimension $d$, its variance remains stable, making it a reliable surrogate for MI maximization. Finally, when repurposed for two-step MI estimation, our method surpasses other variational lower bound estimators across dimensions.

### E.2.4 Comparison of our two-step estimator with other two-step estimators

While our primary contribution is a novel variational lower bound (JSD-LB) that can be directly used for MI maximization, we additionally report results for our two-step estimator (Our MI estim.) alongside other two-step approaches to provide better context. Comparisons with SMILE, KL-DIME, and GAN-DIME are presented in Tables 5 and 6. The results show that our derived two-step MI estimator, which is equivalent to GAN-DIME up to a linear transformation, achieves competitive performance.

### E.2.5 Computational time analysis

A critical characteristic is the computation time. The computational time analysis is developed on a server with CPU "Intel Xeon Platinum 8468 48-Core Processor" and an NVIDIA GPU H100 and reported in Table 7. As in [12], we found that the influence of the MI lower bound objective functions on the algorithm's time requirements is minor.

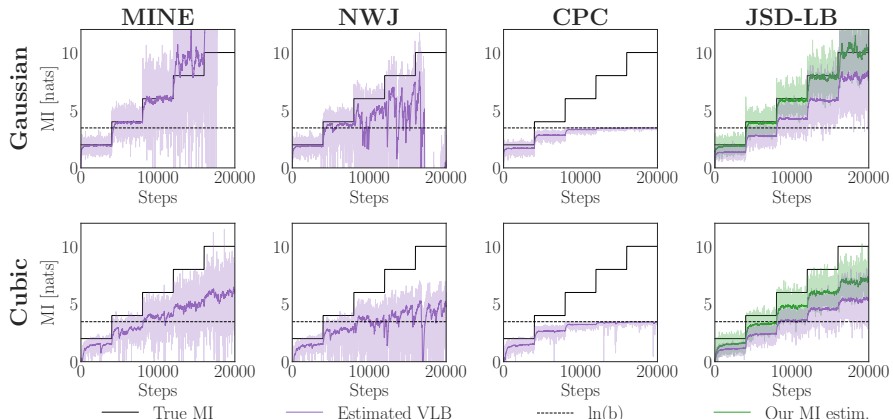

Figure 7: Staircase MI estimation comparison for $d = 5$ and batch size $b = 32$. The Gaussian case is reported in the top row, while the cubic case is shown in the bottom row.

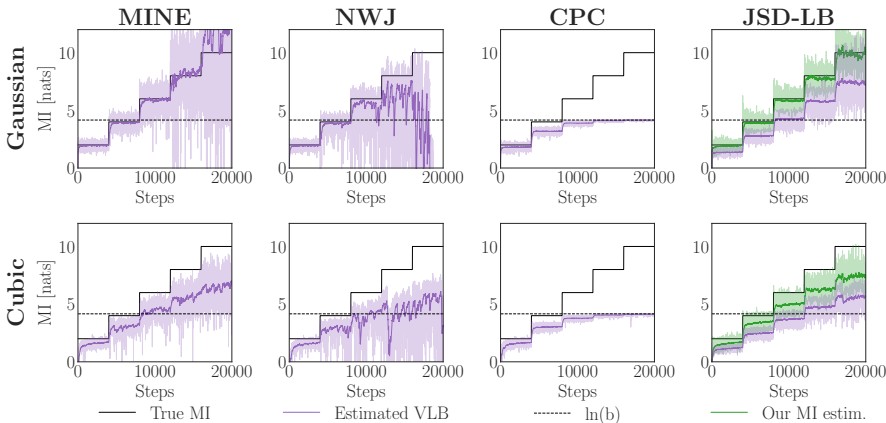

Figure 8: Staircase MI estimation comparison for $d = 5$ and batch size $b = 64$. The Gaussian case is reported in the top row, while the cubic case is shown in the bottom row.

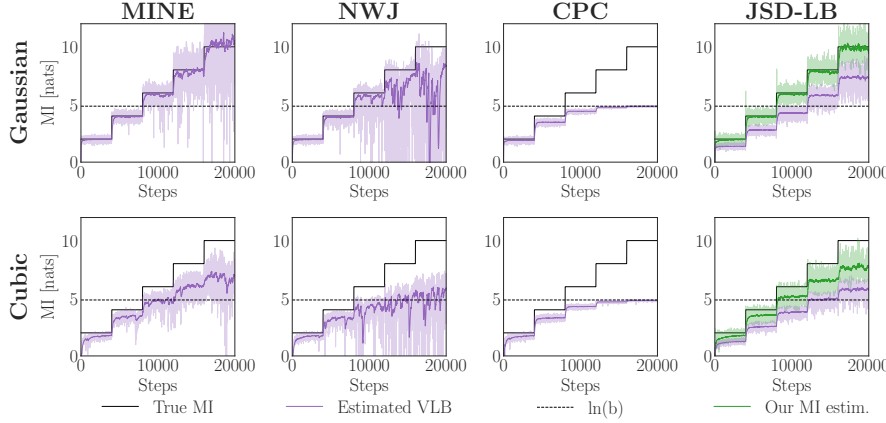

Figure 9: Staircase MI estimation comparison for $d = 5$ and batch size $b = 128$. The Gaussian case is reported in the top row, while the cubic case is shown in the bottom row.

Table 4: Bias, variance, and mean squared error (MSE) of the MI estimators using the joint architecture, when $b = 128$, for the Gaussian setting with varying dimensionality $d \in \{5, 10, 20, 100\}$.

| MI | $d = 5$ | | | | | $d = 10$ | | | | |
|---|---|---|---|---|---|---|---|---|---|---|
| | 2 | 4 | 6 | 8 | 10 | 2 | 4 | 6 | 8 | 10 |
| **a) Bias** | | | | | | | | | | |
| MINE | 0.07 | 0.11 | 0.12 | 0.05 | -1.58 | 0.12 | 0.19 | 0.32 | 0.12 | -1.26 |
| NWJ | 0.09 | 0.19 | 0.8 | 1.77 | $\infty$ | 0.15 | 0.28 | 0.86 | $\infty$ | $\infty$ |
| CPC | 0.2 | 0.81 | 2.11 | 3.89 | 5.85 | 0.25 | 0.91 | 2.21 | 3.94 | 5.86 |
| JSD-LB | 0.64 | 1.22 | 1.75 | 2.22 | 2.62 | 0.77 | 1.55 | 2.33 | 3.07 | 3.75 |
| Our MI estim. | 0.1 | 0.09 | 0.12 | 0.19 | 0.1 | 0.17 | 0.21 | 0.28 | 0.35 | 0.29 |
| **b) Variance** | | | | | | | | | | |
| MINE | 0.05 | 0.14 | 0.41 | 2.05 | 12.23 | 0.06 | 0.15 | 0.49 | 1.63 | 8.87 |
| NWJ | 0.07 | 0.23 | 4.09 | 12.83 | $\infty$ | 0.09 | 0.23 | 1.49 | $\infty$ | $\infty$ |
| CPC | 0.04 | 0.03 | 0.01 | 0.0 | 0.0 | 0.04 | 0.03 | 0.01 | 0.0 | 0.0 |
| JSD-LB | 0.03 | 0.06 | 0.12 | 0.23 | 0.72 | 0.03 | 0.06 | 0.11 | 0.19 | 0.35 |
| Our MI estim. | 0.07 | 0.13 | 0.22 | 0.36 | 0.93 | 0.08 | 0.15 | 0.25 | 0.44 | 0.73 |
| **c) Mean Square Error** | | | | | | | | | | |
| MINE | 0.06 | 0.15 | 0.43 | 2.05 | 14.72 | 0.08 | 0.19 | 0.59 | 1.64 | 10.46 |
| NWJ | 0.07 | 0.27 | 4.74 | 15.98 | $\infty$ | 0.11 | 0.31 | 2.23 | $\infty$ | $\infty$ |
| CPC | 0.08 | 0.69 | 4.45 | 15.15 | 34.23 | 0.1 | 0.87 | 4.89 | 15.5 | 34.36 |
| JSD-LB | 0.45 | 1.55 | 3.17 | 5.16 | 7.57 | 0.63 | 2.47 | 5.56 | 9.62 | 14.43 |
| Our MI estim. | 0.08 | 0.13 | 0.23 | 0.39 | 0.94 | 0.11 | 0.2 | 0.33 | 0.56 | 0.82 |
| MI | $d = 20$ | | | | | $d = 100$ | | | | |
| | 2 | 4 | 6 | 8 | 10 | 2 | 4 | 6 | 8 | 10 |
| **a) Bias** | | | | | | | | | | |
| MINE | 0.19 | 0.38 | 0.8 | 0.82 | 0.7 | 1.0 | 1.36 | 2.61 | 3.24 | 4.0 |
| NWJ | 0.23 | 0.48 | $\infty$ | $\infty$ | $\infty$ | 1.13 | 1.67 | 3.05 | 4.1 | $\infty$ |
| CPC | 0.31 | 1.0 | 2.3 | 4.0 | 5.89 | 0.99 | 1.39 | 2.59 | 4.18 | 5.98 |
| JSD-LB | 0.88 | 1.8 | 2.77 | 3.73 | 4.67 | 1.42 | 2.36 | 3.54 | 4.79 | 6.06 |
| Our MI estim. | 0.26 | 0.39 | 0.52 | 0.64 | 0.65 | 1.1 | 1.22 | 1.67 | 2.22 | 2.83 |
| **b) Variance** | | | | | | | | | | |
| MINE | 0.08 | 0.19 | 0.65 | 1.62 | 5.74 | 0.19 | 0.21 | 0.99 | 1.15 | 1.75 |
| NWJ | 0.12 | 0.27 | $\infty$ | $\infty$ | $\infty$ | 0.14 | 0.17 | 1.19 | 1.81 | $\infty$ |
| CPC | 0.06 | 0.04 | 0.02 | 0.01 | 0.0 | 0.18 | 0.04 | 0.03 | 0.01 | 0.01 |
| JSD-LB | 0.04 | 0.06 | 0.1 | 0.18 | 0.29 | 0.06 | 0.04 | 0.06 | 0.11 | 0.16 |
| Our MI estim. | 0.12 | 0.17 | 0.28 | 0.44 | 0.73 | 0.17 | 0.14 | 0.19 | 0.27 | 0.41 |
| **c) Mean Square Error** | | | | | | | | | | |
| MINE | 0.12 | 0.34 | 1.29 | 2.29 | 6.23 | 1.18 | 2.05 | 7.81 | 11.62 | 17.76 |
| NWJ | 0.17 | 0.5 | $\infty$ | $\infty$ | $\infty$ | 1.41 | 2.95 | 10.46 | 18.61 | $\infty$ |
| CPC | 0.15 | 1.04 | 5.32 | 15.97 | 34.65 | 1.15 | 1.96 | 6.72 | 17.47 | 35.83 |
| JSD-LB | 0.82 | 3.28 | 7.75 | 14.12 | 22.06 | 2.07 | 5.6 | 12.61 | 23.05 | 36.89 |
| Our MI estim. | 0.19 | 0.32 | 0.55 | 0.86 | 1.15 | 1.39 | 1.62 | 2.99 | 5.18 | 8.41 |

Table 5: Bias, Variance, and Mean Square Error of SMILE, KL-DIME, GAN-DIME, and Our MI estimator across different $b$, and MI values for $d = 5$.

| MI | $b = 64$ | | | | | $b = 256$ | | | | |
|---|---|---|---|---|---|---|---|---|---|---|
| | 2 | 4 | 6 | 8 | 10 | 2 | 4 | 6 | 8 | 10 |
| **a) Bias** | | | | | | | | | | |
| SMILE | -0.23 | -0.59 | -0.73 | -0.68 | -0.65 | -0.28 | -0.65 | -0.81 | -0.78 | -0.68 |
| KL-DIME | 0.09 | 0.12 | 0.22 | 0.44 | 0.9 | 0.05 | 0.05 | 0.09 | 0.19 | 0.41 |
| GAN-DIME | 0.1 | 0.1 | 0.13 | 0.24 | 0.09 | 0.06 | 0.05 | 0.07 | 0.12 | 0.01 |
| Our MI estim. | 0.1 | 0.09 | 0.12 | 0.19 | 0.1 | 0.06 | 0.05 | 0.07 | 0.12 | 0.04 |
| **b) Variance** | | | | | | | | | | |
| SMILE | 0.08 | 0.13 | 0.22 | 0.33 | 0.72 | 0.03 | 0.07 | 0.1 | 0.18 | 0.3 |
| KL-DIME | 0.06 | 0.07 | 0.08 | 0.11 | 0.17 | 0.03 | 0.02 | 0.02 | 0.03 | 0.05 |
| GAN-DIME | 0.07 | 0.13 | 0.22 | 0.37 | 0.92 | 0.03 | 0.06 | 0.11 | 0.18 | 0.4 |
| Our MI estim. | 0.07 | 0.13 | 0.22 | 0.36 | 0.93 | 0.03 | 0.06 | 0.1 | 0.2 | 0.43 |
| **c) Mean Square Error** | | | | | | | | | | |
| SMILE | 0.13 | 0.48 | 0.74 | 0.8 | 1.14 | 0.11 | 0.49 | 0.75 | 0.8 | 0.77 |
| KL-DIME | 0.07 | 0.08 | 0.13 | 0.3 | 0.97 | 0.03 | 0.02 | 0.03 | 0.07 | 0.22 |
| GAN-DIME | 0.08 | 0.14 | 0.23 | 0.43 | 0.92 | 0.04 | 0.07 | 0.11 | 0.2 | 0.4 |
| Our MI estim. | 0.08 | 0.13 | 0.23 | 0.39 | 0.94 | 0.04 | 0.07 | 0.11 | 0.22 | 0.43 |

Table 6: Bias, Variance, and Mean Square Error of SMILE, KL-DIME, GAN-DIME, and Our MI estimator across different $b$, and MI values for $d = 20$.

| MI | $b = 64$ | | | | | $b = 256$ | | | | |
|---|---|---|---|---|---|---|---|---|---|---|
| | 2 | 4 | 6 | 8 | 10 | 2 | 4 | 6 | 8 | 10 |
| **a) Bias** | | | | | | | | | | |
| SMILE | -0.08 | -0.24 | -0.25 | -0.24 | -0.16 | -0.17 | -0.38 | -0.43 | -0.39 | -0.28 |
| KL-DIME | 0.24 | 0.36 | 0.62 | 1.0 | 1.57 | 0.16 | 0.2 | 0.34 | 0.58 | 0.92 |
| GAN-DIME | 0.26 | 0.38 | 0.54 | 0.67 | 0.72 | 0.19 | 0.26 | 0.35 | 0.48 | 0.5 |
| Our MI estim. | 0.26 | 0.39 | 0.52 | 0.64 | 0.65 | 0.19 | 0.26 | 0.36 | 0.47 | 0.53 |
| **b) Variance** | | | | | | | | | | |
| SMILE | 0.12 | 0.16 | 0.27 | 0.46 | 0.77 | 0.06 | 0.07 | 0.12 | 0.2 | 0.32 |
| KL-DIME | 0.1 | 0.11 | 0.15 | 0.19 | 0.22 | 0.05 | 0.03 | 0.04 | 0.05 | 0.07 |
| GAN-DIME | 0.12 | 0.17 | 0.26 | 0.41 | 0.73 | 0.06 | 0.08 | 0.12 | 0.2 | 0.36 |
| Our MI estim. | 0.12 | 0.17 | 0.28 | 0.44 | 0.73 | 0.06 | 0.08 | 0.12 | 0.2 | 0.32 |
| **c) Mean Square Error** | | | | | | | | | | |
| SMILE | 0.12 | 0.22 | 0.33 | 0.52 | 0.79 | 0.08 | 0.21 | 0.3 | 0.35 | 0.41 |
| KL-DIME | 0.16 | 0.24 | 0.53 | 1.19 | 2.69 | 0.07 | 0.07 | 0.16 | 0.38 | 0.91 |
| GAN-DIME | 0.19 | 0.32 | 0.55 | 0.86 | 1.25 | 0.09 | 0.15 | 0.24 | 0.43 | 0.61 |
| Our MI estim. | 0.19 | 0.32 | 0.55 | 0.86 | 1.15 | 0.09 | 0.14 | 0.25 | 0.42 | 0.6 |

Table 7: Comparison of the time requirements (in minutes) to complete the 5-step staircase MI ($d = 5$) over the batch size.

| | $b = 32$ | $b = 64$ | $b = 128$ | $b = 256$ | $b = 512$ | $b = 1024$ |
|---|---|---|---|---|---|---|
| MINE | 0.54 | 0.46 | 0.52 | 1.0 | 2.67 | 9.07 |
| NWJ | 0.39 | 0.4 | 0.46 | 0.94 | 2.61 | 9.02 |
| CPC | 0.39 | 0.37 | 0.42 | 0.91 | 2.59 | 8.99 |
| JSD-LB | 0.45 | 0.48 | 0.51 | 0.96 | 2.62 | 9.05 |

