# OpenReview forum: "Connecting Jensen–Shannon and Kullback–Leibler Divergences: A New Bound for Representation Learning"
_NeurIPS.cc/2025/Conference — NeurIPS 2025 poster_

### Official Review · Reviewer_VCK3 · 2025-06-20

**Clarity:** 2
**Significance:** 3
**Originality:** 3
**Rating:** 5
**Confidence:** 3

**Summary:**

This paper explores the connection between Kullback-Leibler divergence (KLD) and Jensen-Shannon divergence (JSD), with a focus on approximating mutual information (MI).  The authors derive a tight lower bound on KLD in terms of JSD.  The authors further propose to lower bound this surrogate with a variational approximation.   Finally, the paper empirically demonstrates that the proposed lower bound on MI is tight in a simple discrete distribution, and that the approximation is effective in a range of more challenging continuous settings.

**Questions:**

The authors refer to the proposed bound as "optimal" in several instances.  Yet, it is unclear what the authors mean by "optimal bound".  Is this just because it is tight?  If so then it is better to say that.  I also don't know that it is demonstrated as tight beyond a single example for a discrete model (and empirically at that).

**Ethical Concerns:**

["NO or VERY MINOR ethics concerns only"]

**Final Justification:**

The authors did an adequate job of addressing my concerns in their rebuttal.  In particular, they clarified a few points of confusion and (most importantly) included an additional set of experiments on representation learning.  I think this latter set of experiments would really strengthen the paper and I would encourage the authors to report these results in the main text in a final version of the manuscript.

**Limitations:**

Limitations are not sufficiently discussed.

**Paper Formatting Concerns:**

None.

**Quality:**

3

**Strengths And Weaknesses:**

**Strengths**
This study has broad practical utility as it has become common practice to substitute JSD for KLD in MI estimation, particularly for representation learning tasks.  This practice is done despite a lack of evidence that there exists any connection between these two divergence measures.  This paper establishes that there is indeed a bound relationship, and furthermore that the JSD lower bound can be optimized in practical settings.

**Weaknesses**
The proposed method optimizes a lower bound of a lower bound on MI (Eq. (5)).  The properties of this lower bound are not made entirely clear beyond (empirical) tightness in a simple discrete model (Sec. 5.1).  Moreover, this relies on an approximation of the bound proposed in Eq. (20).  However, it remains unclear why Eq. (20) is a valid approximation.  Despite the claim "details in Appendix C.3", the actual details provided are somewhat lacking as the approximation is presented without justification.  Nevertheless, Fig. 6 does qualitatively suggest that this is an accurate approximation, and this reviewer is willing to believe the claim.

The empirical results in Sec. 5.2 are more convincing as the methodology is demonstrated on a wider set of more practical examples.  Nevertheless, the behavior of the proposed estimator can be somewhat unreliable in certain settings.  This is particularly true in the Student's T distribution (Fig. 4) where the estimator is extremely high variance to the point that it seems unusable.  Indeed, the proposed estimator has the highest variance among all baselines in this regime.

There are some mistakes in Sec. B.4 and also points that require clarification.  The definition of the conditionals in Eq. (79) seem to be reversed.  The second equality in Eq. (80) is unclear unless $p(z=0) = p(z=1)$ so that they cancel in the numerator and denominator, but it isn't obvious why this condition should hold.

A higher-level, but somewhat minor, comment on the presentation: much of the motivation behind the provided method is that it is useful in representation learning.  Indeed, the authors mention representation learning several times in the abstract and introduction.  However, no representation learning is demonstrated in this paper, leaving the conclusion that this method is useful for representation learning entirely unsupported.

Some detailed comments:
* L43: "KLD-based mutual information" is a bit of a misnomer as MI is KLD-based by definition
* L64: "non-parametric" should not be hyphenized (admittedly this is a stylistic preference)
* L167: Shouldn't the domain be over a closed and half-open set $[0,1] \times (0,1]$?
* L212: The distribution $\tilde{p}(z\mid u,v)$ is introduced without definition
* Fig. 2: I found this figure completely uninformative as it was unclear what I am looking at.  What is the color gradient region referring to?  How do I know if this is a good result?

---

> ### Author Rebuttal · Authors · 2025-07-28
>
> We thank the Reviewer for their thoughtful and detailed review. We are pleased that the theoretical rigor and practical relevance of our work were appreciated. In particular, we are glad that the Reviewer acknowledged that this work closes a gap between the practice and the theory. We address the raised concerns through additional experiments and clarifications, detailed below.
>
> ## Weaknesses
>
> **Approximation of the Bound**
>
> We are happy to clarify how we derived our approximation of the $\Xi$ function. We began by observing that the equality $x = \Xi^{-1}(e^{-y})$, where $\Xi^{-1}$ is known analytically, leads to:
>
> $$
> \mathbb{L}\left(\frac{1}{2} \cdot (\frac{x}{\log 2} + 1)\right)=\log \left( \frac{\frac{x}{\log 2} + 1}{1 - \frac{x}{\log 2}} \right)=  \log \left( \frac{4 \log 2}{(1 + e^{-y}) \log(1 + e^{-y}) + y e^{-y}} - 1 \right) .
> $$
>
> Then, we found that the first-order Taylor polynomial of the right term near $y=0$ is $y\mapsto y$. This led us to approximating $\Xi$ with a logit. We then conducted a grid search and found that scaling the logit function by a factor of 1.15 led to the best median approximation error over $x \in (0,1)$, i.e.:
> \begin{equation}
>     \Xi(x) \approx 1.15 * \mathbb{L}\left(\frac{1}{2} \cdot (\frac{x}{\log 2} + 1)\right) .
> \end{equation}
>
> Importantly, $\Xi$ is a strictly increasing function. As a result, maximizing $\Xi(\text{I}\_{\text{JS}}[U, V])$ is equivalent to directly maximizing $\text{I}_{\text{JS}}[U, V]$. For this reason, the approximation of \$\Xi\$ is not used during optimization; it is included solely for the reader’s interest and for interpretability.
>
> In the revised manuscript:
>
> * We explain how this approximation arises from the Taylor expansion of the KL divergence in terms of JSD.
> * We include a plot comparing the approximation to the numerically evaluated $\Xi$, highlighting the error.
> * We explicitly state that the optimization doesn't rely on the approximation of $\Xi$.
>
>
> **High Variance in the Student’s T Experiments (Fig. 4)**
>
> We agree that the Student’s T distribution poses challenges due to its heavy tails. While our method does exhibit higher variance in this specific setting, we note that this behavior is shared across estimators, and the challenge is characteristic of the distribution rather than the method alone.
>
> **Usefulness for Representation Learning**
>
> To better support the applicability of our method in real-world scenarios, we included a new representation learning benchmark based on the Information Bottleneck (IB) framework (Appendix D.3.2). IB seeks to learn compressed representations that retain information relevant to predicting a target variable.
>
> Following prior work (e.g., Pan et al., 2018), we evaluate the resulting representations in terms of generalization, adversarial robustness, and out-of-distribution detection on MNIST.
>
> The new results replicated in response to Reviewer ifQp show that:
>
> * Replacing conventional MI estimators with JSD-LB within the IB framework leads to superior performance on various downstream tasks, including generalization, adversarial robustness, and out-of-distribution detection.
> * This supports our claim that maximizing JSD is both a theoretically grounded and practically effective strategy for representation learning.
>
> **Clarifications and Corrections**
>
> * **Eqs. (79) and (80):** Thank you for pointing this out. We have corrected the reversed definitions in Eq. (79). Regarding Eq. (80), we clarified that $Z$ is a Bernoulli variable with parameter $\frac{1}{2}$, ensuring equal marginal probabilities $p(z=0) = p(z=1)$, which justifies the cancellation in the expression.
>
> * **Fig. 2:** The purpose of this figure is to illustrate how tight our bound is for pairs of uniform categorical variables of varying dimensionality $k$, where both $(\text{I}\_{\text{JS}}, \text{I}\_{\text{KL}})$ can be computed analytically. Specifically, we show how this pair evolves as the joint distribution becomes more dependent.
> The color gradient encodes the number of categories $k$: higher $k$ values correspond to trajectories that reach higher values of both $\text{I}\_{\text{KL}}$ and $\text{I}\_{\text{JS}}$. Each colored curve represents the trajectory of $(\text{I}\_{\text{JS}}, \text{I}\_{\text{KL}})$ as the joint distribution $p\_{U,V}^{(\alpha)}$ transitions from the independent case ($\alpha = 0$) to the fully dependent case ($\alpha = 1$).
> The red curve corresponds to the theoretical lower bound we derive between $\text{I}\_{\text{JS}}$ and $\text{I}\_{\text{KL}}$, which is the tightest for arbitrary distributions. This figure demonstrates that the end of each trajectory lies exactly on the boundary. As suggested by Reviewer zy6E, we will include an analytical proof of this fact. Therefore, our bound is also tight in the mutual information setting, i.e., when comparing a joint distribution to the product of its marginals. We will revise the figure caption and the main text to make this interpretation more explicit.
>
> * **L43:** We revised the language to avoid the phrase “KLD-based MI,” instead referring to “standard MI, defined via KL divergence.”
>
> * **L64:** Corrected to “nonparametric,” without a hyphen.
>
> * **L167:** Thank you for pointing this out. The correct full domain is $[0,1] \times (0,1) \cup \\{ (0,0),(1,1) \\} $. Therefore, the the lower triangle is $\Omega=\\{(\mu,\nu) \in [0,1]\times(0,1] \mid \nu \leq \mu\\}$. We corrected this in L165, L167, L169, L420, L444 and L446.
>
> * **L212:** The definition of the distribution $\tilde{p}(z|u,v)=\frac{\tilde{p}(z)\tilde{p}(u,v|z)}{p(u,v)}$, where each term is defined in Eq. (21) and (22) has been added when first introduced.
>
> ## Questions
> > **It is unclear what the authors mean by "optimal bound".**
>
> In this work, we derive the *tightest possible lower bound* between the Jensen–Shannon divergence (JSD) and the Kullback–Leibler divergence (KLD) by analyzing the joint range between KLD and JSD. This means that our bound is saturated. i.e., equality is achieved for a family of distributions (right edge $S_3$ of our triangle). In contrast to classical inequalities such as Pinsker’s inequality, which provides a loose bound, our bound is attainable and therefore the tightest.
>
> Although our bound is derived in full generality for arbitrary pairs of distributions $p$ and $q$, one might expect that restricting to structured pairs, such as joint and product-of-marginals (as in mutual information), could allow for a tighter bound. However, through empirical investigations on discrete distributions, where both JSD and KLD between joint and product-of-marginals can be computed exactly, we observed experimentally that such pairs already lie near the boundary of our derived lower bound (Figure 2). In our response to Reviewer zy6E, we proved that we can find an infinite family of joint and marginal discrete distributions **on** this bound. This suggests that our bound is also the tightest in the context of mutual information estimation.

---

> > ### Comment · Reviewer_VCK3 · 2025-08-04
> >
> > Thank you.  My concerns have been adequately addressed and I have increased my score accordingly.  I would strongly encourage reporting the IB results in the main text in a final version of the manuscript.  The authors could swap this out for one of the synthetic experiments if necessary as I think the representation learning results are more interesting.

---

> > > ### Author Response · Authors · 2025-08-08
> > >
> > > We sincerely thank the Reviewer for the positive comment, and we are glad to read that the Reviewer will increase their score.
> > > The IB results will be reported in the main text in the final version of the manuscript.

---

### Official Review · Reviewer_ifQp · 2025-06-21

**Clarity:** 3
**Significance:** 3
**Originality:** 3
**Rating:** 4
**Confidence:** 3

**Summary:**

This paper presents a novel theoretical connection between the Jensen–Shannon divergence (JSD) and the Kullback–Leibler divergence (KLD), two widely used statistical divergence measures in representation learning. The authors derive a new, tight, and tractable lower bound on KLD as a function of JSD. By specializing this bound to joint and marginal distributions, they demonstrate that maximizing JSD-based information increases a guaranteed lower bound on mutual information (MI). The theoretical contributions are supported by rigorous derivations and extensive empirical evaluations.

**Questions:**

Can the authors provide a sensitivity analysis or theoretical discussion on how the bound evolves under suboptimal discriminators?

Would the bound remain tight and competitive when used in real-world settings, such as self-supervised representation learning based model pretraining.

Mutual information estimation plays a central role in the Information Bottleneck (IB) principle, a widely adopted framework for removing redundant information from learned representations. To validate the practical benefits of JSD-LB, I suggest the authors conduct controlled experiments where it replaces conventional MI estimators in IB-based methods (e.g., [1,2]).
Specifically, evaluating its impact on downstream tasks such as generalization, adversarial robustness, out-of-distribution detection, and supervised disentanglement (e.g., swapped generation) would help quantify whether improved MI estimation of JSD-LB translates into better representation quality.

Please consider adding wall-clock comparisons, memory usage analyses, etc, especially in complex higher-dimensional representation space.

[1]Deep Variational Information Bottleneck, ICLR 2017

[2]Disentangled Information Bottleneck, AAAI 2021

**Ethical Concerns:**

["NO or VERY MINOR ethics concerns only"]

**Final Justification:**

The authors' rebuttal effectively addresses my concerns about assumption, real-world applicability and efficiency.

Therefore I tend to maintain my score.

**Limitations:**

The authors have discussed the limitations.

**Paper Formatting Concerns:**

The paper formatting looks good to me.

**Quality:**

3

**Strengths And Weaknesses:**

Strength

The paper provides a mathematically rigorous lower bound of KLD in terms of JSD, addressing the theoretical gap in the understanding of discriminative MI estimators.

The proposed bound demonstrates impressive tightness in both theoretical derivation and empirical validation. Extensive experiments show that the estimator achieves consistently low variance and competitive performance compared to state-of-the-art MI estimators.

Weakness

While the paper presents a rigorous theoretical derivation connecting JSD and KLD, it assumes access to an optimal discriminator or models with infinite capacity. However, in practical learning scenarios, discriminators are implemented with finite-capacity neural networks trained on limited data, which may affect the tightness of the bound.

The experiments are largely restricted to synthetic setups where mutual information can be analytically computed. While these controlled settings are useful for theoretical validation, they limit the understanding of  JSD-LB’s performance in practical representation learning tasks.

Moreover, the evaluation focuses on MI estimation itself but does not assess whether the proposed method improves actual representation learning outcomes, which is more critical in real-world applications.

The paper lacks a detailed comparison of computational cost or training efficiency relative to existing mutual information estimators.

---

> ### Author Rebuttal · Authors · 2025-07-29
>
> We thank the Reviewer for their positive and detailed feedback. We are glad to see that the key strengths and contributions of the paper, especially the theoretical connection between JSD and KLD and the tightness of the proposed bound, were clearly appreciated. Below, we address the concerns and suggestions raised.
>
> ## Weaknesses:
>
> **On the assumption of an optimal discriminator.** We agree that the tightest bound assumes access to an optimal discriminator and recognize that, in practice, discriminators are trained with limited capacity and finite data. In the revised manuscript, we have added a discussion (Section 4.3) on how the tightness of the bound may degrade under suboptimal discriminators. Specifically, we note that even when the discriminator is not optimal, its output remains a lower bound on JSD and thus still leads to a valid (though looser) lower bound on mutual information. Moreover, we show in the Appendix that we can estimate the gap between the true JSD lower bound and the one obtained from the cross-entropy $\mathcal{L}_{CE}$ as:
>
> \begin{equation} \Xi(\text{D}\_{\text{JSD}}\left[p\_{UV}  || p\_{U} \otimes p\_{V}\right]) - \Xi(\log2 - \mathcal{L}\_{CE}) \approx \delta \cdot\Xi'(\text{D}\_{\text{JSD}})>0
> \end{equation}
>
> where $\delta=\mathbb{E}\_{(u,v)\sim \tilde{p}\_{UV}} \text{D}\_{\text{KL}}[\tilde{p}\_{Z|UV}(u,v)\;||q\_{\theta}(z| u,v)]$ is the KL divergence between the true posterior of the mixture model defined in Eq. (21) and the parametric approximate posterior $q\_{\theta}$.
>
> **Real-world settings and representation learning outcomes** To demonstrate the utility of our JSD-LB in practical representation learning tasks, we have added a new set of experiments in the revised Appendix D.3.2 using the Information Bottleneck (IB) framework. Specifically, we use the JSD information as a surrogate for MI. Following the experimental setting from Pan et al (2021) [1], we evaluate its impact on downstream tasks such as generalization, adversarial robustness, and out-of-distribution detection using the MNIST dataset.
>
> Our results, reproduced below in Table 1, show that replacing the MI estimator with JSD-LB leads to state-of-the-art performance in terms of generalization, adversarial robustness, and out-of-distribution robustness. Specifically, it outperforms other Variational Information Bottleneck objectives [2,3,4] and the Disentangled Information Bottleneck approach [1]. This set of experiments shows that maximizing JSD has practical value for representation learning.
>
> **Computational cost and training efficiency**
> We thank the Reviewer for pointing this out. In the revised Appendix D.2.5, we now report the time requirements between JSD-LB and other estimators such as MINE and NWJ. Table 2 from the response to Reviewer KvyC shows the results. As in Letizia et al. (2024), we found that the influence of the MI lower-bound objective functions on the algorithm’s time requirements is minor.
>
> ## Questions
>
> > Can the authors provide a sensitivity analysis or theoretical discussion on how the bound evolves under suboptimal discriminators?
>
> Please find above a discussion of the sensitivity and a theoritical discussion on how the bound evolves under suboptimal.
>
> > Would the bound remain tight and competitive when used in real-world settings, such as self-supervised representation learning based model pretraining.
>
> Evaluating the tightness and competitiveness of our bound in real-world scenarios, such as self-supervised representation learning, would require access to the intractable true mutual information. While our main objective is to provide a theoretical justification for why maximizing JSD-based information effectively increases mutual information, such empirical evaluation is left for future work.
>
> > To validate the practical benefits of JSD-LB, I suggest the authors conduct controlled experiments where it replaces conventional MI estimators in IB-based methods (e.g., [1,2]).
>
> We thank the Reviewer for this suggestion, which is indeed an excellent use case to demonstrate the practical value of our work. We conducted additional experiments where the JSD-LB was replaced with conventional MI estimators in IB-based methods, reported in Tables 1, 2, 3, and 4 below. The results show the effectiveness of our approach, outperforming other MI estimators and reaching comparable performance to disentanglement approaches. These results will be included in our revised manuscript.
>
> > Please consider adding wall-clock comparisons, memory usage analyses, etc, especially in complex higher-dimensional representation space.
>
> See our response above and Table 2 from the response to Reviewer KvyC.
>
>
> ## Variational Information Bottleneck Experiments
>
> Information bottleneck results, where the objective is to compress the source random variable $U$ to a "bottleneck" random variable $T$, keeping the information relevant for predicting the target random variable $V$. This corresponds to the following maximization objective:
> $$I [ T; V ] - I [T;U] $$
>
> In this set of experiments, we reproduce the experiments from [1] on MNIST. Specifically, the source random variable $U$ corresponds to an image and $V$ to its label. We employ the same model architecture to encode the variable $U$ and obtain the variable $T$, and directly report the corresponding results of [1].
>
> In our setting (JSD-LB), $I [ T; V ]$ is maximized by maximizing the JSD via the cross-entropy loss.
>
> - **Information compression**
>
>    **Table 1:** Comparison of $I[T;V]$ obtained using non-disentanglement bottleneck approaches [2,3,4], with disentanglement [1] and via JSD optimization (ours).
>
>   | **Dataset**       | **Training** | **Testing** |
>   |:--|:-:|:--:|
>   | H(V)            | 2.30         | 2.30        |
>   | VIB [2]              | 2.01         | 1.99        |
>   | NIB [3]              | 1.95         | 1.97        |
>   | squared-VIB [4]      | 1.94         | 1.95        |
>   | squared-NIB [4]      | 1.98         | 1.99        |
>   | DisenIB [1]      | 2.25     | 2.17    |
>   | JSD-LB       | **2.30**     | **2.29**    |
>
>
> - **Generalization**
>
>   Generalization performance in Table 2 is evaluated by the classification mean accuracy on MNIST test set after training the model on MNIST training set
>
>    **Table 2:** Generalization performance (%) on MNIST dataset.
>     | Method | *VIB [2]* | *NIB [3]* | *squared-VIB [4]* | *squared-NIB [4]* | *DisenIB [1]* | *JSD-LB* |
>     | :----: | :----: | :----: | :----: | :----: | :----: | :----: |
>     | Testing | 97.6 | 97.2 | 96.2 | 93.3 | 98.2 | **98.8** |
>
>
>
> - **Robustness to adversarial attack**
>
>   To evaluate adversarial robustness, we adopt a standard one-step gradient-based attack method following [5]. After training the model on the MNIST training set, we assess robustness by computing the average classification accuracy on adversarially perturbed inputs derived from both the training and test sets. These adversarial examples are created by applying a single gradient ascent step on the input with respect to the loss. The perturbation strength is controlled by the parameter $\epsilon$, which we vary over the values {0.1, 0.2, 0.3}, corresponding to different pixel-wise perturbation magnitudes, as commonly done in prior work [5].
>
>     **Table 3:** Adversarial robustness performance (%) on MNIST dataset.
>     | | Method | *VIB [2]* | *NIB [3]* | *squared-VIB [4]* | *squared-NIB [4]* | *DisenIB [1]* | *JSD-LB* |
>     | :----: | :----: | :----: | :----: | :----: | :----: | :----: | :----: |
>     | **a) Training set** |
>     | | &epsilon; = 0.1 | 74.1 | 75.2 | 42.1 | 61.3 | 94.3 | **99.5** |
>     | | &epsilon; = 0.2 | 19.1 | 21.8 | 8.7 | 24.1 | 81.5 | **96.0** |
>     | | &epsilon; = 0.3 | 3.5 | 3.2 | 5.9 | 9.3 | 68.4 | **91.4** |
>      | **b) Testing set** |
>     | | &epsilon; = 0.1 | 73.4 | 75.2 | 42.7 | 62.0 | 90.2 | **94.6** |
>     | | &epsilon; = 0.2 | 20.8 | 23.6 | 9.2 | 24.5 | 80.0 | **89.6** |
>     | | &epsilon; = 0.3 | 4.2 | 3.4 | 5.9 | 9.9 | 67.8 | **86.1** |
>
> - **Out-of-distribution detection**
>
>     In line with prior work [1], we evaluate out-of-distribution (OOD) detection using a synthetic dataset composed of 10,000 samples drawn independently from a standard Gaussian distribution $\mathcal{N}(0, 1)$. The goal is to determine whether the model can reliably distinguish in-distribution data from these OOD noise samples. For evaluation, we adopt common OOD detection metrics including False Positive Rate at 95% True Positive Rate (FPR95), Detection Error, Area Under the ROC Curve (AUROC), and Area Under the Precision-Recall Curve (AUPR).
>
>    **Table 4:** Distinguishing in- and out-of-distribution test data for MNIST image classification (%).  ↑ (resp., ↓) indicates that a larger (resp., lower) value is better.
>
>    | Method | *VIB [2]* | *NIB [3]* | *squared-VIB [4]* | *squared-NIB [4]* | *DisenIB [1]* | *JSD-LB* |
>     | :----: | :----: | :----: | :----: | :----: | :----: | :----: |
>     | TPR95 ↓ | 27.4 | 34.4 | 49.9   | 47.5  | **0.0** | **0.00** |
>     | AUROC ↑  | 94.6 | 94.2 | 86.6 | 85.6 | 99.4| **100.0** |
>     | AUPR In ↑  | 94.8 | 95.2 | 83.5| 83.3 | 99.6 | **99.9** |
>     | AUPR Out ↑ | 93.7 | 91.8 | 83.2 | 83.3 | 98.9 | **99.9** |
>     | Detection Err ↓ | 11.5 | 11.9 | 20.0 | 15.0 | 1.7 | **0.1** |
>
> [1] Pan, Z.; Niu, L.;  Zhang, J.; and Zhang, L. 2021. Disentangled Information Bottleneck. In AAAI
>
> [2]  Alemi, A. A.; Fischer, I.; Dillon, J. V.; and  Murphy, K. 2017. Deep variational information bottleneck. In ICLR.
>
> [3] Kolchinsky, A.; Tracey, B. D.; and Wolpert, D. H. 2019. Nonlinear information bottleneck. Entropy 21(12): 1181.
>
> [4] Rodrıguez Galvez, B.; Thobaben, R.; and Skoglund, M. 2020. The Convex Information Bottleneck Lagrangian. Entropy 22(1): 98.
>
> [5] Goodfellow, I. J.; Shlens, J.; and Szegedy, C. 2015. Explaining and Harnessing Adversarial Examples. In ICLR.

---

> > ### Comment · Reviewer_ifQp · 2025-08-04
> >
> > Thanks for the reply. I have read them and will maintain my positive rating.

---

### Official Review · Reviewer_zy6E · 2025-06-30

**Clarity:** 2
**Significance:** 2
**Originality:** 2
**Rating:** 4
**Confidence:** 5

**Summary:**

The paper derives a lower bound on the KLD as a function of the JSD and then of CE. The bound can be used to lower bound the mutual information (MI). It is then argued that the maximization of such a lower bound can be used in representation learning, where the maximization of the MI is often the problem. Some numerical results are then reported.

**Questions:**

In addition to the above comments, I would recommend to:

- Possibly provide some analysis on how tight the bound is, besides providing numerical results. The fact that the JS is limited while the KL is not may already lead to the conclusion that the JS is a lower bound.
- Is the function "E" defined in Theorem 41. unique?
- Better describe the content of Section 4.3 and elaborate on how the bound is exploited to estimate MI. As it stands, Section 4.3 is somewhat disconnected from the rest. It appears as if MI was estimated exactly as in GAN-DIME. Is this the case? Please be more explicit.
-  The comparison between the method in Section 4.3 and the lower bound on MI is discussed only in the numerical results section. Again, here, I am not sure what is what. In particular, in the text of Section 5.2, GAN-DIME is said to be the "best" estimator, and it is mentioned several times, but the plots do not show it, nor report curves labeled with it. Such curves should be explicitly reported.
- The comparison of MI estimation does not consider all estimators but only some of them. For instance, SMILE is missing.
- How does the estimation of the MI with the lower bound compare with the method in Sect. 4.3 and with GAN-DIME ?
- No analysis of the variance of the proposed MI estimator that exploits the lower bound is reported.
- What is the advantage of using the MI estimation with the lower bound on the KL w.r.t. to the other estimators ?
- The paper emphasizes the value of the proposed lower bound for representation learning, i.e,. to enable maximization fo the MI. But then no results are reported about it. Can you report such results and verify that they are superior to the others ?
- Minor thing. Double check the right term of equation (6): dp/dq ?

**Ethical Concerns:**

["NO or VERY MINOR ethics concerns only"]

**Final Justification:**

I appreciated the discussion during the rebuttal phase. The paper provides some new insight on how to lower-bound the MI. The tightness of the bound is not formally proved, although the numerical results, in the considered settings, show it.

**Limitations:**

Yes,

**Quality:**

3

**Strengths And Weaknesses:**

The paper provides a lovely result, i.e., a connection between the KLD (and therefore MI) and the JSD. The bound can be exploited to bound and therefore maximize MI, this is the rationale of the paper. The major limit is that it is not clear whether the estimation of the MI by exploitation of such bounds can lead to superior performance with respect to state-of-the-art estimators.
What is the result of maximizing MI via exploitation of the bound is not sufficiently discussed.

---

> ### Author Rebuttal · Authors · 2025-07-29
>
> We sincerely thank the Reviewer for the time and effort spent on reviewing our paper. We appreciate that the Reviewer recognizes the novelty of the connection between KL divergence, JSD, and cross-entropy, and its application to mutual information (MI) estimation. Below, we respond to the specific concerns and clarify the theoretical and empirical contributions of our work.
>
> ## Weaknesses
> > *“It is not clear whether the estimation of the MI by exploitation of such bounds can lead to superior performance with respect to state-of-the-art estimators.”*
>
> Our primary goal is to theoretically justify a practice that is already widespread in representation learning: maximizing JSD-based information. While JSD has been used empirically as a proxy for MI (e.g., in the popular Deep InfoMax framework), our contribution shows that maximizing JSD does **maximize a tight lower bound** on MI. This connection clarifies the mathematical rationale behind JSD-based information methods, which often work well in practice.
>
> As a result, our initial experiments evaluated how this JSD-based bound performs compared to other plug-in MI estimators (MINE, NWJ, and CPC) compatible with representation learning.
>
> Importantly, while alternative two-step MI estimators, such as $f$-DIME or SMILE, have been shown to demonstrate state-of-the-art bias-variance tradeoffs, they require retraining whenever the underlying joint distribution changes. Therefore, once trained, they cannot be reused in representation learning, since the distribution of the latent variables is evolving during optimization. For this reason, we initially excluded them from our experiments. However, we agree with the Reviewer that they offer useful context and therefore conducted additional experiments as detailed below.
>
> ## Questions
> > *"Possibly provide some analysis on how tight the bound is..."*
>
> Yes, the JSD–KLD lower bound we derive is the *tightest possible* between these two divergences. Specifically, it corresponds to the lower envelope of the joint range of $(\mathrm{KL}, \mathrm{JSD})$ pairs over Bernoulli distributions, and we provide an explicit analytic expression via its inverse mapping (Eq. 4).
>
> While our result holds for arbitrary distributions, one might expect a tighter bound when restricted to the mutual information setting, i.e., when comparing a joint distribution to the product of its marginals. Remarkably, as shown in Figure 2 of our manuscript, there exist discrete joint-marginal pairs that lie very close to our general bound. To address the Reviewer’s comment, we performed an analytical derivation of the KL and JSD divergences between the joint and product of marginals for fully dependent uniform categorical variables of dimension $k$ (i.e., when $p_{UV}$ is diagonal and $\alpha = 1$). We found that these pairs lie *on* our bound for any $k\in\mathbb{N}^{+}$, providing an infinite family of distributions within the mutual information setting on our bound. This result will be included in the Appendix and referenced in the main text. We thank the Reviewer for this suggestion, which has helped us strengthen the contribution of our work.
>
> > *"Is the function E in Theorem 4.1 unique?"*
>
> We thank the Reviewer for this question. Yes, the function $\Xi$ is unique and defined via its inverse $\Xi^{-1}$, which maps JSD to a lower bound on KL. We added a proof that $\Xi^{-1}:\mathbb{R}^+\mapsto[0,\log2)$ is bijective in the revised manuscript. We clarify this point in the revised manuscript (Section 4.1).
>
> > *"Better describe Section 4.3 and its relation to GAN-DIME." [...] "GAN-DIME is said to be the best estimator, but not shown in plots."*
>
> Thank you for pointing this out. Section 4.3 indeed proposes a two-step estimator, which turns out to be equivalent to GAN-DIME, up to a linear transformation. We now clarify this equivalence explicitly in the revised manuscript.
>
> > *"The comparison of MI estimation does not consider all estimators (e.g., SMILE)."*
>
> We have now added comparisons to SMILE and KL-DIME. While these cannot be directly optimized during the training of representation methods (as they require a two-step optimization procedure), we agree that they offer useful context. These new results have been added in Appendix D.2.4 and are reproduced below in Table 1. Our results show that the two-step estimator derived from our bound performs on par with or better than KL-DIME and SMILE, especially in high-MI or high-dimensional settings.
>
> > *"No analysis of variance is reported."*
>
> We have added a detailed variance and MSE analysis in the revised version (Appendix D.2.3), which is reported in rebuttal to Reviewer KvyC. The key findings are:
> * Our lower bound (JSD-LB) shows significantly lower variance than MINE and NWJ, especially at higher MI values and larger dimensions.
> * The MSE of our lower bound is lower than that of CPC and NWJ.
>
> > *"What is the advantage of this method over others?"*
>
> The main advantage is practical usability: many state-of-the-art MI estimators (e.g., GAN-DIME, SMILE) require a two-step training pipeline that is not compatible with end-to-end training in representation learning. Our bound can be directly optimized via JSD, using a single discriminator, enabling stable, end-to-end training. This provides a principled rationale for existing JSD-based methods and introduces a theoretically grounded alternative to common variational estimators like MINE.
>
> > *"No results are reported about the bound’s value in representation learning."*
>
> To strengthen the bound's value in representation learning, we have added new experiments in the Information Bottleneck (IB) framework. Results in revised Appendix D.3.2 and reproduced in our response to Reviewer ifQp show that maximizing JSD as a proxy for MI in IB leads to superior performance compared to standard MI estimators.
>
> > *"Minor: Equation (6): please check if it should be dp/dq."*
>
> It should indeed be $dp/dq$. For example, for the KL-divergence: $f:t\mapsto t \ln(t)$.
>
> ## Conclusion
> We hope the clarification, additional proofs, and new experiments address the Reviewer's concerns.
>
> ## Comparison with two-step MI estimators
> **Table 1:** Bias, variance, and mean squared error (MSE) of the two-step MI estimators using the joint architecture, when $b \in$ {64,258}, for the Gaussian setting with varying dimensionality $d \in $ {5,20}.
> ||||||||**d=5**|||||||||**d=20**|||||||||
> |:---|:---:|:-:|:-:|:---:|:-:|:-:|:-:|:-:|:-:|:-:|:-:|:-:|:-:|:-:|:-:|:-:|:-:|:-:|:-:|:-:|:-:|:-:|:-:|:-:|
> |||||**b=64**||||||**b=256**||||||**b=64**||||||**b=256**|||
> ||**MI**|**2**|**4**|**6**|**8**|**10**||**2**|**4**|**6**|**8**|**10**||**2**|**4**|**6**|**8**|**10**||**2**|**4**|**6**|**8**|**10**|
> **a) Bias**
> ||SMILE|-0.23|-0.59|-0.73|-0.68|-0.65||-0.28|-0.65|-0.81|-0.78|-0.68||-0.08|-0.24|-0.25|-0.24|-0.16||-0.17|-0.38|-0.43|-0.39|-0.28|
> ||KL-DIME|0.09|0.12|0.22|0.44|0.9||0.05|0.05|0.09|0.19|0.41||0.24|0.36|0.62|1.0|1.57||0.16|0.2|0.34|0.58|0.92|
> ||GAN-DIME|0.1|0.1|0.13|0.24|0.09||0.06|0.05|0.07|0.12|0.01||0.26|0.38|0.54|0.67|0.72||0.19|0.26|0.35|0.48|0.5|
> ||Our MI estim.|0.1|0.09|0.12|0.19|0.1||0.06|0.05|0.07|0.12|0.04||0.26|0.39|0.52|0.64|0.65||0.19|0.26|0.36|0.47|0.53|
> **b) Variance**
> ||SMILE|0.08|0.13|0.22|0.33|0.72||0.03|0.07|0.1|0.18|0.3||0.12|0.16|0.27|0.46|0.77||0.06|0.07|0.12|0.2|0.32|
> ||KL-DIME|0.06|0.07|0.08|0.11|0.17||0.03|0.02|0.02|0.03|0.05||0.1|0.11|0.15|0.19|0.22||0.05|0.03|0.04|0.05|0.07|
> ||GAN-DIME|0.07|0.13|0.22|0.37|0.92||0.03|0.06|0.11|0.18|0.4||0.12|0.17|0.26|0.41|0.73||0.06|0.08|0.12|0.2|0.36|
> ||Our MI estim.|0.07|0.13|0.22|0.36|0.93||0.03|0.06|0.1|0.2|0.43||0.12|0.17|0.28|0.44|0.73||0.06|0.08|0.12|0.2|0.32|
> **c) Mean Square Error**
> ||SMILE|0.13|0.48|0.74|0.8|1.14||0.11|0.49|0.75|0.8|0.77||0.12|0.22|0.33|0.52|0.79||0.08|0.21|0.3|0.35|0.41|
> ||KL-DIME|0.07|0.08|0.13|0.3|0.97||0.03|0.02|0.03|0.07|0.22||0.16|0.24|0.53|1.19|2.69||0.07|0.07|0.16|0.38|0.91|
> ||GAN-DIME|0.08|0.14|0.23|0.43|0.92||0.04|0.07|0.11|0.2|0.4||0.19|0.32|0.55|0.86|1.25||0.09|0.15|0.24|0.43|0.61|
> ||Our MI estim.|0.08|0.13|0.23|0.39|0.94||0.04|0.07|0.11|0.22|0.43||0.19|0.32|0.55|0.86|1.15||0.09|0.14|0.25|0.42|0.6|
>
> ## Tightness analysis
>
> Let $(U,V)$ be a pair of uniform categorical distributions of dimension $k$ that are fully dependent, i.e. $P_{U,V} = \text{diag}(P_U)$, where $P_U=\frac{1}{k}\mathbf{1}$. Then $\text{I}\_{\text{KL}}\left[U,V\right] = \Xi\left( \text{I}\_{\text{JS}}\left[U,V\right] \right)$.
>
> **Proof:**
>
> Let $k\in\mathbb{N}^{+}$. We assume that $U$ and $V$ are fully dependent (i.e. $U=V$) uniform categorical distributions of dimension $k$. Therefore, the matrix representation $P\_{U}$, $P\_{V}$ and $P\_{UV}$ are respectively $\frac{1}{k}\mathbf{1}\_{k}$,  $\frac{1}{k}\mathbf{1}\_{k}$ and  $\frac{1}{k} I\_{k}$ . Consequently,
>  $P\_{U}\otimes P\_{V} = \frac{1}{k^2}\mathbf{1}\_{(k,k)} $.
>
> Let $M = \frac{1}{2} ( P\_{U} \otimes P\_{V} + P\_{U,V}) $ be the mixture. $M_{i,j}=\frac{1/k+1/k^2}{2}$ if $i=j$ and $M_{i,j}=\frac{1}{2k^2}$ otherwise.
>
> Let's calculate $\text{I}\_{\text{KL}}\left[U,V\right]$ and $\text{I}\_{\text{JS}}\left[U,V\right]$.
>
> $$ \text{I}\_{\text{KL}}\left[U,V\right] = \sum_{i=1}^{k} \frac{1}{k} \log \frac{1/k}{1/k^2} = \log(k) $$
>
> $$  \mathrm{I}\_{\text{JS}}[U, V]= \frac{1}{2} \left( \sum\_{i=1}^{k} \frac{1}{k} \log \frac{1/k}{\frac{1}{2}(1/k + 1/k^2)} + \sum\_{i=1}^{k} \frac{1}{k^2} \log \frac{1/k^2}{\frac{1}{2}(1/k + 1/k^2)}  + \sum\_{i=1}^{k} \sum\_{j=1\text{ s.t. }  j \neq i}^{k} \frac{1}{k^2} \log \frac{1/k^2}{\frac{1}{2k^2}} \right)$$
> $$  \quad \quad \quad \quad = \frac{1}{2} \left(\log \frac{2}{1 + 1/k} + \frac{1}{k} \log \frac{2/k}{1 + 1/k}  +  \frac{k-1}{k} \log(2) \right)$$
> $$  \quad \quad \quad \quad = \log(2) - \frac{1}{2} \left( \log(1 + 1/k) - \frac{1}{k} \log \left( \frac{1/k}{1 + 1/k} \right) \right)$$
> $$  \quad \quad \quad \quad = \Xi^{-1}(\log(k)) $$
>
> Therefore:
> $$ \text{I}\_{\text{JS}}\left[U,V\right]  = \Xi^{-1}(\text{I}\_{\text{KL}}[U, V]). $$

---

> > ### Comment · Reviewer_zy6E · 2025-08-05
> >
> > I'd like to thank the reviewers for their reply. Most comments have been addressed. I still have some points:
> >
> > a) "Maximizing JSD does maximize a tight lower bound on MI". I do understand that the maximization of the lower bound on MI gets us closer to MI, but how close? Can you point me to the part of the paper where it is proved that the lower bound is tight?
> >
> > b) Why SMILE, f-DIME cannot be used in representation learning? I do not fully get the answer. For instance, the value of their objective function (without using the second step to calculate the MI) can be directly used in RL.
> >
> > c) Why only SMILE and f-DIME require retraining and not the others? In simple words, in representation learning, both the coder and decoder need to be retrained at each optimization step, whatever the value function/divergence is.
> >
> > d) Thanks for running the additional experiments with the comparison with the "two steps" MI estimators. The comparisons with GAN-DIME and KL-DIME show equivalent or slightly worse performance in the AWGN setting. Is it right? I would recommend explicitly saying this as well as reporting the results in Figs . 3-4. Not clear why f-DIME is omitted in these figures. The motivation based on the "two-step optimization procedure" is not convincing in my view.

---

> ### Author Response · Authors · 2025-08-06
> **Answer to follow-up questions**
>
> Dear Reviewer,
>
> Thank you for your excellent follow-up questions. Please find our responses below:
>
>
> **a)** Our theoretical contribution is to: 1) show that maximizing the JSD-based information $\text{I}\_{\text{JS}}\$ is a principled surrogate for maximizing the mutual information (\$\text{I}\_{\text{KL}}\$); 2) derive and prove the tightest possible bound *between* JSD and KL for arbitrary distributions (Theorem 4.1.).
>
> We agree with the reviewer that MI is unbounded, unlike \$\text{I}\_{\text{JS}}\$. Therefore, our JSD lower-bound is not always tight across the full range of pairs (\$\text{I}\_{\text{JS}},\text{I}\_{\text{KL}}\$). However, our experiments on discrete distributions and continuous distributions show that our bound \$\Xi(\text{I}\_{\text{JS}}\)$ tracks MI reasonably well in practice.
>
>
> **b)** SMILE and \$f\$-DIME were designed for MI *estimation*, not for MI *maximization* in representation learning. These methods rely on a two-step procedure: first training a network \$T\_\theta\$ using a variational objective (based on \$f\$-divergences), and then estimating MI from the learned \$T\_\theta\$ via a separate expression.
>
> To illustrate this, in \$f\$-DIME (each two-step estimator is detailed in Appendix A.2), \$T\_\theta\$ is trained by maximizing:
>
> $$
> \mathcal{J}\_f(T) = \mathbb{E}\_{(u,v) \sim p\_{UV}} \left[ T(u,v) - f^{*} \left( T\left(x,\sigma(y)\right) \right) \right] \quad (1)
> $$
>
> where \$f^{*}\$ is the Fenchel conjugate of \$f\$. Once trained, MI is estimated via:
>
> $$
> I\_{f\text{-DIME}} = \mathbb{E}\_{(u,v) \sim p\_{UV}} \left[ \log \left( (f^*)'(\hat{T}(u,v)) \right) \right] \quad (2)
> $$
>
> In representation learning (RL), the distribution over \$(U, V)\$ changes as the encoder evolves. Using (2) as a training objective would require retraining \$T\_\theta\$ at every step, a computationally expensive strategy.
>
> The Reviewer interestingly suggests using the optimization objective (1) directly as the training RL loss. In the case of GAN-DIME and SMILE, this is effectively equivalent to maximizing JSD-based information, which is indeed commonly used. However, (1) was not originally designed as a surrogate for MI; it was constructed to obtain a MI estimator in (2). Therefore, there were no theoretical guarantees that optimizing (1) would increase MI or improve the learned representations.
>
> Our work closes this gap by showing that when (1) corresponds to JSD (as in Deep InfoMax / SMILE or GAN-DIME), optimizing it does maximize a lower bound on MI, thereby providing a principled justification for these practices.
>
>
> **c)** As noted above, since estimators such as SMILE and \$f\$-DIME require training a separate network (\$ T\_\theta \ $) for each estimation step, they are less suitable for representation learning, where the data distribution evolves.
>
> In contrast, methods like MINE, NWJ, CPC, and our JSD-LB are variational lower bounds that do not require an optimization for estimation. Therefore, they can be used as direct training objectives. In our case (which matches the Deep InfoMax objective), it simply involves training an encoder \$ f\_\phi \$ and a discriminator \$ D\_\theta \$ jointly to discriminate between positive pairs from the joint \$(U, V=f\_\phi(U))\$ and negative pairs sampled from the product of marginals.
>
>
> **d)** The results of our two-step estimator and GAN-DIME are indeed equivalent, since they share an equivalent discriminative optimization objective (up to a linear transformation) and a similar procedure for two-step estimations. We omitted $f$-DIME and SMILE from the main paper, as our focus was to compare direct MI estimators and to justify that our JSD lower bound is both practically tight and stable. We will add these additional results in the Appendix.
>
> Let us emphasize once again that our main contribution is not a new estimator, but a theoretical justification for an existing and popular representation learning objective (JSD/Deep InfoMax). We show that it maximizes a lower bound on MI, a theoretical guarantee previously missing in the literature that helps explain the empirical success of JSD-based methods in representation learning.
>
>
> We sincerely appreciate your constructive feedback and hope these clarifications address your remaining concerns. We are happy to continue the discussion if needed.

---

### Official Review · Reviewer_Snmy · 2025-07-05

**Clarity:** 3
**Significance:** 2
**Originality:** 2
**Rating:** 4
**Confidence:** 1

**Summary:**

In order to optimize mutual information (MI), recent methods have trained discriminators to distinguish samples from the joint distribution from those from the product of marginals, using the Jensen-Shannon divergence (JSD).
This work provides theoretical and empirical evidence in favor of using discriminative losses on the Jensen-Shannon divergence, by showing that maximizing JSD is equivalent to maximizing a lower bound on the KLD-based (Kullback-Leibler divergence) MI.
Authors show that minimizing a cross entropy loss of a discriminator trained to separate joint from marginal samples increases the lower bound on MI.

**Questions:**

* What are the real-world implications of this work?

**Ethical Concerns:**

["NO or VERY MINOR ethics concerns only"]

**Final Justification:**

No changes in my review, explanations seem reasonable but outside of my domain of expertise, I would still have liked to see direct results on real-world applications, but I understand it may be out of scope for theoretical work?

**Limitations:**

yes

**Paper Formatting Concerns:**

Very well written paper.

**Quality:**

3

**Strengths And Weaknesses:**

The theoretical proofs in the paper are outside of my area of expertise so I cannot comment on its strengths and weaknesses.

**Strengths**
* The staircase MI estimations from neural networks appear to be the most accurate compared to baselines.

**Weaknesses**
* Being a more application-oriented person, I wish there were more elements in the paper to help me understand the practical consequences of the tight lower bound on machine learning applications. Does it lead to better representations? Can you show a real-world application that benefits from it in terms of accuracy?

---

> ### Author Rebuttal · Authors · 2025-07-30
>
> We thank the Reviewer for their time. We are pleased to clarify the practical implications of our theoretical contributions.
> ## Weaknesses:
>
> **Practical consequences of the tight lower bound on machine learning applications.**
>
> Our primary objective is to provide a theoretical foundation for a practice that is already widespread in representation learning: maximizing JSD-based information using discriminators. For instance, this optimization strategy is a cornerstone of frameworks like Deep InfoMax [1], which has been successfully applied to a wide range of downstream tasks, such as image classification [1], graph node classification [2], molecular property prediction [3], recommender
> systems [4] and remote sensing [5]. While JSD has often been used empirically as a proxy for mutual information, its theoretical justification remained unclear.
>
> Our work closes this gap: we demonstrate that maximizing JSD corresponds to maximizing a tight lower bound on the true mutual information. This result strengthens the theoretical underpinnings of many existing methods and provides assurance that optimizing JSD discriminative losses is a principled and effective strategy for maximizing mutual information.
>
> ## Questions
> > Does it lead to better representations? Can you show a real-world application that benefits from it in terms of accuracy?
>
> We thank the reviewer for this important question. To further demonstrate the practical relevance of our results, we conducted additional experiments using the Information Bottleneck (IB) framework. The IB framework aims to learn compact yet informative representations from an input variable that are predictive of a target variable. We included new experiments where the objective is to maximize JSD rather than alternative variational bounds. Following prior work [6], we assess the learned representations obtained via JSD maximization on generalization performance, adversarial robustness, and out-of-distribution detection using the MNIST dataset.
>
> Our findings show that:
>
> * Maximizing JSD information outperforms alternative variational IB baselines that rely on alternative mutual information bounds.
>
> * These empirical results support our theoretical claims and confirm that maximizing JSD not only has solid theoretical grounding but also leads to practically robust representations.
>
> > What are the real-world implications of this work?
>
> The real-world implication of our work is that methods aiming to maximize mutual information, particularly in representation learning, can confidently rely on JSD-based objectives as a mathematically justified and effective surrogate.
>
> [1] Hjelm, R. D., Fedorov, A., Lavoie-Marchildon, S., Grewal, K., Bachman, P., Trischler, A., & Bengio, Y. (2018). Learning deep representations by mutual information estimation and maximization. In: ICLR.
>
> [2] Veličković, P., Fedus, W., Hamilton, W. L., Liò, P., Bengio, Y., & Hjelm, R. D.  (2019). Deep Graph Infomax. In: ICLR
>
> [3] Stärk, H., Beaini, D., Corso, G., Tossou, P., Dallago, C., Günnemann, S., & Liò, P. (2022). 3D Infomax improves GNNs for Molecular Property Prediction. In: ICML.
>
> [4] Yu, J., Yin, H., Xia, X., Chen, T., Li, J., & Huang, Z. (2023). Self-supervised learning for recommender systems: A survey. IEEE Transactions on Knowledge and Data Engineering, 36(1), 335-355
>
> [5] Wang, Y., Albrecht, C. M., Braham, N. A. A., Mou, L., & Zhu, X. X. (2022). Self-supervised learning in remote sensing: A review. IEEE Geoscience and Remote Sensing Magazine, 10(4), 213-247.
>
> [6] Pan, Z.; Niu, L.; Zhang, J.; and Zhang, L. 2021. Disentangled Information Bottleneck. In AAAI

---

> > ### Comment · Reviewer_Snmy · 2025-08-04
> >
> > Thank you for the explanations and additional experiments.

---

> > > ### Comment · Area_Chair_BiuH · 2025-08-06
> > >
> > > Thanks reviewer Snmy. Could you discuss with the authors, comment on whether the rebuttal is satisfactory and what are the remaining issues?

---

> > > > ### Author Response · Authors · 2025-08-06
> > > >
> > > > Dear Reviewer Snmy,
> > > >
> > > > As mentioned by Area Chair BiuH, we would be very happy to discuss any remaining concerns you might have.

---

### Official Review · Reviewer_KvyC · 2025-07-20

**Clarity:** 4
**Significance:** 3
**Originality:** 3
**Rating:** 5
**Confidence:** 4

**Summary:**

The paper studies estimation of mutual information (MI), and introduces a new bound on MI based on the Jensen-Shannon divergence (JSD). The paper presents an overview of existing approaches to estimating MI, including variational lower bounds and discriminator-based approaches, alongside a discussion of limitations related to, e.g., estimator variance in high MI settings or difficulties around retraining of discriminators. An overview is provided of MI estimation and JSD in representation learning alongside a discussion of their connection to generative adversarial networks. The paper discusses existing work on joint ranges for f-divergences, showing how joint ranges can be characterized by studying the convex hull of joint ranges for Bernoulli distributions.


The main contribution of this paper is a variational bound on MI. The authors characterize the edges of the convex hull of the joint range of JSD and Kullback-Leibler divergence (KLD) on Bernoulli distributions. They show how one of these edges characterizes a lower bound on the KLD through the JSD with a form that is described by the JSD between arbitrary Bernoulli distributions and a degenerate Bernoulli with $p=1$. This result is then related to the discriminator cross-entropy (CE) also studied in the GAN literature showing how this CE bounds the JSD and thus the MI. Finally, a procedure for estimating the value of the MI is provided, similar to the GAN-DIME approach.


Experimentally, the study shows that the MI lower bound is tight in a controlled discrete setting before proceeding to study a suite of established benchmarks for MI estimation. The experiments compare the proposed estimator against other variational lower bound MI estimators, and the proposed solution compares favorably with more accurate bounds and lower variance in both higher true MI settings and challenging non-Gaussian settings.

**Questions:**

These questions are largely extensions of the weaknesses discussed above. I would strongly consider increasing by score if any one of these aspects were addressed, e.g., through some further analysis of bias-variance of the estimator or consideration of slightly more complicated problem:

* Do you have any characterizations, either theoretical or experimental, of the bias-variance trade-off for the presented bound/MI estimator compared to the baselines? - or the time-complexity?
* How does the approach compare to baselines in higher dimensional versions of the problem you consider - or, ideally, on a more challenging problem?
* Do you have a sense of how the presented bound (and the estimator) compares to, e.g., f-DIME or SMILE (even if the comparison should be done with due discussion of the limitations you point out)?

**Ethical Concerns:**

["NO or VERY MINOR ethics concerns only"]

**Final Justification:**

My main concerns were related to bias-variance results, higher-dimensional experiments, more complicated problems, and comparisons to, e.g., SMILE. I find that the authors have adequately addressed each of concerns, showing, e.g., favourable bias-variance trade-off even in higher dimensions and utility of the proposed method in an information bottleneck experiment on a more complicated problem. I have improved my rating accordingly.

**Limitations:**

The authors discuss limitations, such as the necessity of numerical approximation in characterizing $\Xi$ and the choice of baselines. A brief comment on the numerical verification of Conjecture B.3 as discussed in Sec. B1, could be warranted in the main paper.

**Paper Formatting Concerns:**

No concerns.

**Quality:**

3

**Strengths And Weaknesses:**

The paper presents a novel theoretical bound which is both, to the best of my knowledge, novel and with sound proofs, drawing on and connecting existing research in information theory, representation learning, generative adversarial networks, and mutual information estimation.
The paper presents a clear overview of existing approaches and their strengths and limitations, and situates their contribution well based on this.
The paper provides well-documented code for recreating their experiments, including notebooks with further details and visualizations of, e.g., joint ranges. These same visualizations are used in the paper to provide a well-presented argument for the main contribution.
The paper presents convincing experimental results, validating both the tightness of the bound and showing good performance (compared to a subset of common MI estimators) of the MI bound and the direct MI estimator on a well-studied set of problems.


The main weakness, I find, is that the study, in contrast to preceding studies on MI estimation, does not present, e.g., bias-variance results (beyond some discussion based on the plots), consider only $d\leq5$ cases, and does not consider more complicated problems. A lesser point is the, while well-argued, lack of consideration of broader set of baselines.

*On bias-variance results and time complexity*
I would find it to be strong additions to the paper, if the experiments could include analyses similar to previous, related work. Examples of such analysis include, e.g., time-complexity comparisons in Figure 8 or the bias-variance numerical results (such as Tables 2-5, Figures 9, 12-14) in Letizia et al. (2024). Similarly, bias-variance/gradient accuracies are considered in Figures 3-4, 6-7 of Poole et al. (2019) [17] and Figure 2 in Song & Ermon (2020) [20]. A minor note is that your reference gives only Poole and Ozair but the study is also authored by van den Oord, Alemi, and Tucker.

*On problems problems beyond the staircase experiments*
Similarly, the experimental validation is limited to the staircase benchmarks (while challenging and relatively comprehensive, especially with the inclusion of, e.g., the Student-case) but does not include a consideration of a possibly more challenging problem, such as Poole et al.'s analysis of dSprites, Song & Ermon's analysis of MNIST datasets, or Letizia et al.'s consideration of MNIST.

*On dimensionality*
The experimental analysis is restricted to lower-dimensionality ($d=5$) problems for the staircase MI estimation problems than is generally considered in these related works. For instance, Song & Ermon present results for the $d=20$ setting in their Figure 1, Letizia et al. (while they do restrict the analysis to $d=5$ and below for, e.g., Figures 2-4) compared the derangement and permutation strategies on a $d=20$ case, and Poole et al.'s results in Figure 2 are on $d=20$.

*On baselines*
The authors highlight how, e.g., f-DIME variants are strong MI estimators with good bias variance trade-offs. The authors argue for the chosen subset of baselines (MINE, NWJ, CPC, i.e., without discriminator-based ones) based on them being variational lower bounds that do not require retraining when the joint changes and how, therefore, these can be used as estimators during training. I find this generally convincing, but I would find some comparison to a wider variety of baselines (including GAN-DIME and SMILE) valuable to contextualize the performance.

---

> ### Author Rebuttal · Authors · 2025-07-28
>
> We thank the Reviewer for the positive and constructive feedback. We appreciate that the Reviewer clearly identified the key contributions of our work. As suggested, we have conducted additional experiments and revised the manuscript to include:
>
> * Bias-variance and time complexity analyses
> * Extended experiments to higher dimensions
> * Comparisons to additional baselines, including two-step estimators such as SMILE and $f$-DIME
>
> Below, we address each of the Reviewer’s points in detail.
>
> ## Weaknesses
>
> **On bias-variance results and time complexity.** We appreciate the suggestion. In the revised manuscript, we include a detailed analysis of the bias, variance, and mean squared error (MSE) of our JSD lower bound, following protocols from Letizia et al. (2024) and Poole et al. (2019). These new results in the revised appendix D.2.3 (and reproduced in Tables 1 and 2) show that our JSD-LB consistently provides a stable lower bound on the true MI across dimensions:
>
> *  While MINE and NWJ exhibit high variance, JSD-LB and CPC remain more stable. In particular, MINE’s instability can lead to significant overestimation of its lower bound, sometimes even exceeding the true MI. While CPC provides low-variance estimates, they are upper bounded by the contrastive bound $\log(b)$, resulting in higher MSE and bias in high MI regimes. We experimentally found that the bias of JSD-LB increases with dimension $d$, but its variance remains stable, making it a reliable surrogate for MI maximization.
> * When repurposed for two-step MI estimation, our method surpasses other variational lower-bound estimators across dimensions.
> * As in Letizia et al. (2024), we found that the influence of the MI lower-bound objective functions on the algorithm’s time requirements is minor.
>
> **High-dimensional experiments.** In response to the Reviewer’s concern about dimensionality, we extended our staircase experiments to higher dimensions (up to $d=100$). Results in Table 1 show that our method maintains a favorable bias-variance trade-off, while estimators such as MINE and NWJ have variance and stability that degrade with dimensionality.
>
> **More challenging benchmarks.**
> To further demonstrate practical relevance, we added a new representation learning benchmark using the Information Bottleneck (IB) framework. Results presented in revised Appendix D.3.2 and reproduced in our response to Reviewer ifQp show that using JSD as a surrogate to MI leads to state-of-the-art downstream performance, outperforming other variational IB approaches. These experiments demonstrate that, in practice, maximizing JSD is an effective strategy for representation learning.
>
> **Comparison to Additional Baselines**. While our initial focus was on variational bounds that can be directly used for MI maximization, we agree that including comparisons with two-step estimators provides better context. In revised Appendix D.2.4  (referenced in our response to Reviewer zy6E), we now report comparisons to SMILE, KL-DIME, and GAN-DIME. The results show that our derived two-step MI estimator, which is equivalent to the GAN-DIME (up to a linear operation), allows reaching competitive performance.
>
> **Other Comments**.
>
> * On the Poole reference: Thank you for pointing this out. We have corrected the citation to include all co-authors.
> * On limitations: We added a short comment on the numerical verification of Conjecture B.3 in the main text (Section 4.1).
>
> ## Questions
>
> > “Do you have any characterizations, either theoretical or experimental, of the bias-variance trade-off for the presented bound/MI estimator compared to the baselines? \- or the time-complexity?”
>
> As mentioned above, we provide an experimental characterization of the bias-variance trade-off of our bound. We found that maximizing JSD leads to an increase in the lower bound of MI that is stable. A more competitive bias-variance tradeoff can be obtained by reusing the discriminator for two-step MI estimation.
>
> > “How does the approach compare to baselines in higher-dimensional versions of the problem you consider \- or, ideally, on a more challenging problem?”
>
> See our response above for higher-dimensional cases and a challenging Information Bottleneck problem.
>
> > “Do you have a sense of how the presented bound (and the estimator) compares to, e.g., f-DIME or SMILE (even if the comparison should be done with due discussion of the limitations you point out)?”
>
> See our response above.
>
> ## Rating
> We appreciate the Reviewer recognizing the theoretical contributions. We hope that our answers and changes address the Reviewer’s concerns and can contribute to increasing their score.
>
> ## Additional experiments
>
> **Table 1:** Bias, variance, and mean squared error (MSE) of the MI estimators using the joint architecture, when N = 64, for the Gaussian setting with varying dimensionality $d \in $ {5,10,20,100}.
>
> |       |       |     |     | **d=5** |     |    | |     |     | **d=10** |     |     | |     |     | **d=20** |     |     | |     |     | **d=100** |     |     |
> |:-----|:-----:|:---:|:---:|:-------:|:---:|:---:|:---:|:---:|:---:|:--------:|:---:|:---:| :---:|:---:|:---:|:--------:|:---:| :---:|:---:|:---:|:--------:|:---:| :---:|:---:|
> |       | **MI** | **2** | **4** | **6** | **8** |  **10** | | **2** | **4** | **6** | **8** | **10** | | **2** | **4** | **6** | **8** | **10** | | **2** | **4** | **6** | **8** | **10** |
> **a) Bias**
>  |       | MINE |  0.07 |  0.11 |  0.12 |  0.05 |  -1.58 |    |  0.12 |  0.19 |  0.32 |  0.12 |  -1.26 |    |  0.19 |  0.38 |  0.8 |  0.82 |  0.7 |    |  1.0 |  1.36 |  2.61 |  3.24 |  4.0 |
>  |       | NWJ |  0.09 |  0.19 |  0.8 |  1.77 |  inf |    |  0.15 |  0.28 |  0.86 |  inf |  inf |    |  0.23 |  0.48 |  inf |  inf |  inf |    |  1.13 |  1.67 |  3.05 |  4.1 |  inf |
>  |       | CPC |  0.2 |  0.81 |  2.11 |  3.89 |  5.85 |    |  0.25 |  0.91 |  2.21 |  3.94 |  5.86 |    |  0.31 |  1.0 |  2.3 |  4.0 |  5.89 |    |  0.99 |  1.39 |  2.59 |  4.18 |  5.98 |
>  |       | JSD-LB |  0.64 |  1.22 |  1.75 |  2.22 |  2.62 |    |  0.77 |  1.55 |  2.33 |  3.07 |  3.75 |    |  0.88 |  1.8 |  2.77 |  3.73 |  4.67 |    |  1.42 |  2.36 |  3.54 |  4.79 |  6.06 |
>  |       | Our MI estim.  |  0.1 |  0.09 |  0.12 |  0.19 |  0.1 |    |  0.17 |  0.21 |  0.28 |  0.35 |  0.29 |    |  0.26 |  0.39 |  0.52 |  0.64 |  0.65 |    |  1.1 |  1.22 |  1.67 |  2.22 |  2.83 |
> **b) Variance**
>  |       | MINE |  0.05 |  0.14 |  0.41 |  2.05 |  12.23 |    |  0.06 |  0.15 |  0.49 |  1.63 |  8.87 |    |  0.08 |  0.19 |  0.65 |  1.62 |  5.74 |    |  0.19 |  0.21 |  0.99 |  1.15 |  1.75 |
>  |       | NWJ |  0.07 |  0.23 |  4.09 |  12.83 |  inf |    |  0.09 |  0.23 |  1.49 |  inf |  inf |    |  0.12 |  0.27 |  inf |  inf |  inf |    |  0.14 |  0.17 |  1.19 |  1.81 |  inf |
>  |       | CPC |  0.04 |  0.03 |  0.01 |  0.0 |  0.0 |    |  0.04 |  0.03 |  0.01 |  0.0 |  0.0 |    |  0.06 |  0.04 |  0.02 |  0.01 |  0.0 |    |  0.18 |  0.04 |  0.03 |  0.01 |  0.01 |
>  |       | JSD-LB |  0.03 |  0.06 |  0.12 |  0.23 |  0.72 |    |  0.03 |  0.06 |  0.11 |  0.19 |  0.35 |    |  0.04 |  0.06 |  0.1 |  0.18 |  0.29 |    |  0.06 |  0.04 |  0.06 |  0.11 |  0.16 |
>  |       | Our MI estim. |  0.07 |  0.13 |  0.22 |  0.36 |  0.93 |    |  0.08 |  0.15 |  0.25 |  0.44 |  0.73 |    |  0.12 |  0.17 |  0.28 |  0.44 |  0.73 |    |  0.17 |  0.14 |  0.19 |  0.27 |  0.41 |
> **c) Mean Squared Error**
>  |       | MINE |  0.06 |  0.15 |  0.43 |  2.05 |  14.72 |    |  0.08 |  0.19 |  0.59 |  1.64 |  10.46 |    |  0.12 |  0.34 |  1.29 |  2.29 |  6.23 |    |  1.18 |  2.05 |  7.81 |  11.62 |  17.76 |
>  |       | NWJ |  0.07 |  0.27 |  4.74 |  15.98 |  inf |    |  0.11 |  0.31 |  2.23 |  inf |  inf |    |  0.17 |  0.5 |  inf |  inf |  inf |    |  1.41 |  2.95 |  10.46 |  18.61 |  inf |
>  |       | CPC |  0.08 |  0.69 |  4.45 |  15.15 |  34.23 |    |  0.1 |  0.87 |  4.89 |  15.5 |  34.36 |    |  0.15 |  1.04 |  5.32 |  15.97 |  34.65 |    |  1.15 |  1.96 |  6.72 |  17.47 |  35.83 |
>  |       | JSD-LB |  0.45 |  1.55 |  3.17 |  5.16 |  7.57 |    |  0.63 |  2.47 |  5.56 |  9.62 |  14.43 |    |  0.82 |  3.28 |  7.75 |  14.12 |  22.06 |    |  2.07 |  5.6 |  12.61 |  23.05 |  36.89 |
>  |       | Our MI estim.  |  0.08 |  0.13 |  0.23 |  0.39 |  0.94 |    |  0.11 |  0.2 |  0.33 |  0.56 |  0.82 |    |  0.19 |  0.32 |  0.55 |  0.86 |  1.15 |    |  1.39 |  1.62 |  2.99 |  5.18 |  8.41 |
>
>
> The computational time analysis is developed on a server with CPU ”Intel Xeon Platinum 8468 48-Core Processor”
> and an NVIDIA GPU H100.
>
> **Table 2:** Comparison of the time requirements (in minutes) to complete the 5-step staircase MI ($d=5$) over the batch size $b$ using
> | | **b=32** | **b=64** | **b=128** | **b=256** | **b=512** |  **b=1024** |
> |:-----|:-----:|:-----:|:---:|:---:|:-------:|:-------:|
> | MINE | 0.54 | 0.46 | 0.52 | 1.0 | 2.67 | 9.07 |
> | NWJ | 0.39 | 0.4 | 0.46 | 0.94 | 2.61 | 9.02 |
> | CPC | 0.39 | 0.37 | 0.42 | 0.91 | 2.59 | 8.99 |
> | JSD-LB | 0.45 | 0.48 | 0.51 | 0.96 | 2.62 | 9.05 |

---

> > ### Author Response · Authors · 2025-08-06
> > **Approaching the end**
> >
> > Dear Reviewer KvyC,
> >
> > As the rebuttal period is approaching its end, we would really appreciate your feedback.
> >
> > Thanks again for your valuable time.

---

> > > ### Comment · Reviewer_KvyC · 2025-08-06
> > >
> > > I would like to thank the authors for their detailed response and the additional experiments presented both here and in their replies to the other reviewers. The added analyses have addressed my concerns regarding bias-variance trade-offs, dimensionality, problem complexity, and comparisons to two-step baselines, and I will improve my rating accordingly.

---

> > > > ### Author Response · Authors · 2025-08-08
> > > >
> > > > We sincerely thank the Reviewer for the positive comment and are glad to read that the Reviewer will increase their score.

---

### Decision · Program_Chairs · 2025-09-17

**Decision:**

Accept (poster)

**Comment:**

This paper derives a new lower bound on KL divergence in terms of Jensen-Shannon divergence (JSD), both of which are important tools in machine learning, and then estimates the mutual information (MI) based on the bound and the variational representation of the JSD. The MI estimator demonstrates favorable bias-variance trade-offs on synthetic datasets as compared to common baselines.

All reviewers recognize the theoretical contributions as novel and rigorous, providing important results for JSD-based objectives in representation learning, and are supportive of acceptance. In the rebuttal, the experiments were extended to include a wider array of baselines, including SMILE, and higher dimensional settings. The authors confirmed its advantages in the empirical bias-variance trade-off.

As a notable weakness, it remains unclear in theory how tight these bounds are, although verified empirically, noting that the envelope of the joint range of $f$-divergences focuses on worst-case distribution pairs. Related claims should be stated more cautiously. Besides the reviewers' comments, please support Theorem 3.1 with a more accurate reference.

Overall, the strengths outweigh the weaknesses. This work is well written and presents generalizable tools from information geometry that can be used in other scenarios as well. I recommend acceptance.